# TRAINING MULTI-LAYER OVER-PARAMETRIZED NEURAL NETWORK IN SUBQUADRATIC TIME

## ABSTRACT

In the recent years of development of theoretical machine learning, over-parametrization has been shown to be a powerful tool to resolve many fundamental problems, such as the convergence analysis of deep neural network. While many works have been focusing on designing various algorithms for over-parametrized network with one-hidden layer, multiple-hidden layers framework has received much less attention due to the complexity of the analysis, and even fewer algorithms have been proposed. In this work, we initiate the study of the performance of second-order algorithm on multiple-hidden layers over-parametrized neural network. We propose a novel algorithm to train such network, in time subquadratic in the width of the neural network. Our algorithm combines the Gram-Gauss-Newton method, tensor-based sketching techniques and preconditioning.

## 1 INTRODUCTION

Deep neural networks have been playing a central role in both practical (such as computer vision (LeCun et al., 1998; Krizhevsky et al., 2012; Szegedy et al., 2015; He et al., 2016), natural language processing (Collobert et al., 2011; Devlin et al., 2018), automatic driving system, game playing (Silver et al., 2016; 2017) ) and theoretical machine learning community (Li & Liang (2018); Jacot et al. (2018); Du et al. (2019b); Allen-Zhu et al. (2019a;b); Du et al. (2019a); Song & Yang (2019); Brand et al. (2021); Zou et al. (2018); Cao & Gu (2019); Lee et al. (2019a); Liu et al. (2020; 2021); Chen et al. (2021)) . In order to analyze the dynamics of neural networks and obtain provable guarantees, using over-parametrization has been a growing trend.

In terms of understanding the convergence behavior of over-parametrized networks, most of the attentions have been directed to the study of *first-order method* such as gradient descent or stochastic gradient descent. The widespread use of first-order method is explained, to a large degree, by its computational efficiency, since computing the gradient of the loss function at each iteration is usually cheap and simple, let alone with its compatibility with random sampling-based method such as mini-batch. One of the major drawbacks of first-order methods is their convergence rate is typically slow in many non-convex settings ($\mathrm{poly}(n, L, \log(1/\epsilon))$ iterations, where $n$ is the number of training samples, $L$ is the number of layers and $\epsilon$ is the precision of training), e.g., deep neural network with ReLU activation, as shown in Allen-Zhu et al. (2019a)), which is often the case of a deep over-parameterized neural network.

Second-order method (which employs the information of the Hessian matrix) on the other hand enjoys a much faster convergence rate (only $\log(1/\epsilon)$ iterations (Zhang et al., 2019), but not $\mathrm{poly}(n) \log(1/\epsilon)$ iterations) and exploits the local geometry of the loss function to overcome the pathological curvature issues that are critical in first-order method. Another clear advantage of second-order method over first-order method is it does not require the tuning of learning rate. The expense of using second-order method is its prohibitive *cost per iteration*, as it is imperative to *invert* a dynamically-changing Hessian matrix or equivalently, solving a regression task involving Hessian. Given any weight matrix of size $m \times m$ ($m$ is the width of network), its Hessian matrix has size $m^2 \times m^2$, which makes any naive implementation of second-order algorithm takes at least $O(m^4)$ time since one needs to write down the Hessian. This explains the scarcity of deploying large-scale second-order method in non-convex setting, such as training deep neural networks, in contrast to their popular presence in convex optimization setting (Vaidya (1989); Daitch & Spielman (2008); Lee et al. (2015); Cohen et al. (2019); Lee et al. (2019b); Jiang et al. (2020b;a); Song & Yu (2021)).

Recent works (Cai et al. (2019); Zhang et al. (2019)) improved the practicality of second-order method on training deep networks and presented algorithms to train one-hidden layer over-parametrized networks with smooth activation functions. Specifically, they achieve a running time of $O(mn^2)$ per iteration. Their methods are essentially variants of Gauss-Newton method combined with using neural tangent kernels (Jacot et al. (2018)) to prove the convergence. By using the idea of randomized linear algebra (Clarkson & Woodruff (2013); Woodruff (2014)) and a clever sketching matrix as a preconditioner, Brand et al. (2021) further improves the running time to $\widetilde{O}(mn)$ per iteration.

However, all of these algorithms are for training a shallow network with one-hidden layer and fall short on *deep* networks — First, it is not clear that their algorithms can be generalized to multi-layer setting, due to the presence of gradient vanishing or exploding. In the seminal work of Allen-Zhu et al. (2019a), they showed that as long as the networks are over-parametrized, first-order methods such as gradient descent and stochastic gradient descent won't encounter such problem. But does this still hold for second-order method? Can we *provably* show that second-order method has a good performance when training deep over-parametrized networks? Second, even the fastest of them (Brand et al. (2021)) would incur a running time of $\widetilde{O}(m^2 nL)$ per iteration, which seems unavoidable due to the size of intermediate weight matrices is $m \times m$.

In this work, we take the first step to tame the beast — We propose a second-order method that achieves *subquadratic cost per iteration* with respect to $m$, and show that it has *linear convergence rate* in training deep over-parametrized neural networks. We emphasize the importance of obtaining subquadratic algorithm, since in multi-layer settings, the network width is typically much larger than in one-hidden layer setting ($m \geq n^8 L^{12}$, (Zou & Gu, 2019)).

Our work can be decomposed into two parts: algorithmically and analytically. From an algorithmic perspective, our method builds upon a variant of Gauss-Newton method (Björck (1996)) called *Gram-Gauss-Newton* method (Cai et al. (2019); Brand et al. (2021)). In order to achieve a feasible running time, we exploit two features of the gradient, which is the key ingredient to form the Jacobian matrix: 1). The gradient is low rank (rank $n$). 2). The gradient can be formulated as the outer product of two vectors. From an analytical perspective, our work is inspired by Allen-Zhu et al. (2019a). In contrast to their proof which is a straightforward analysis of the gradient, we make use of the multi-layer neural tangent kernels (Du et al. (2019a)) and establish a connection between a Gram matrix we compute at each iteration and the NTK matrix.

**Our Contributions.** We summarize our technical contributions below.

- We develop an analytical framework for the convergence behavior of second-order method on training multi-layer over-parametrized neural network. To facilitate the analysis, we exploit the equivalence between neural tangent kernels and our over-parametrized network.
- We design a second-order algorithm to train such networks, and achieve a cost per iteration of $\widetilde{o}(m^2)$. Our algorithm makes use of Gram-Gauss-Newton method, tensor-based sketching techniques, and data structures that maintains a low rank representation efficiently.
- By combining fast tensor algebra techniques and sketching-based preconditioning, we devise an algorithm to efficiently solve a regression problem where the involved matrix has its rows being tensor product of vectors.

## 1.1 Our Result

Our main result can be summarized in the following three theorems, with one analyzing the convergence behavior of a general Gram-based optimization framework, one designing an efficient algorithm to realize this second-order optimization scheme, and the other is a novel algorithm to solve tensor-based regression in high precision and fast, which is a key step in our second-order method.

Throughout this paper, we will use $n$ to denote the number of training data points, $d$ to denote the dimension of input data points, $m$ to denote the width of the network and $L$ to denote the number of layers of the network. We use $f_t \in \mathbb{R}^n$ to denote the prediction of neural network at time $t$.

Our first theorem demonstrates the fast convergence rate of our algorithm.

**Theorem 1.1** (Convergence, informal version of Theorem F.19)**.** *Suppose the width of the neural network satisfies $m \geq \mathrm{poly}(n, L)$, then there exists an algorithm (Algorithm 1) such that, over*

*the randomness of initialization of the network and the algorithm, with probability at least $1 - e^{-\Omega(\log^2 m)}$, we have*

$$\|f_{t+1} - y\|_2 \leq \frac{1}{2}\|f_t - y\|_2,$$

*where $f_t \in \mathbb{R}^n$ is the the prediction produced by neural network at time $t$.*

The above theorem establishes the linear convergence behavior of our second-order method, which is a standard convergence result for second-order method, as well as the same behavior as in one-hidden layer over-parametrized networks (Brand et al. (2021)). However, compared to one-hidden layer case, our analysis is much more sophisticated since we have to carefully control the probability so that it does not blow up exponentially with respect to the number of layers.

The next theorem concerns the *cost per iteration* of our second-order algorithm.

**Theorem 1.2** (Runtime, informal version of Theorem B.1). *There exists a randomized algorithm (Algorithm 1) that trains a multi-layer neural network of width $m$ with the cost per training iteration being*

$$O(m^{2-\Omega(1)}).$$

We improve the overall training time of multi-layer over-parametrized networks with second-order method from $\mathcal{T}_{\text{init}} + T \cdot O(m^2)$ to $\mathcal{T}_{\text{init}} + T \cdot o(m^2)$, where $\mathcal{T}_{\text{init}}$ is the initialization time of training, typically takes $O(m^2)$. As we have argued before, multi-layer over-parametrized networks require $m$ to be in the order of $n^8$, hence improving the cost per iteration from quadratic to subquadratic is an important gain in speeding up training. Its advantage is even more evident when one seeks a *high precision solution*, and hence the number of iterations $T$ is large.

We highlight that it is non-trivial to obtain a subquadratic running time per iteration: If not handled properly, computing the matrix-vector product with weight matrices will take $O(m^2)$ time! This means that even for first-order methods such as gradient descent, it is not clear how to achieve a subquadratic running time, since one has to multiply the weight matrix with a vector in both forward evaluation and backpropagation. In our case, we have also a Jacobian matrix of size $n \times m^2$, so forming it naively will cost $O(nm^2)$ time, which is prohibitively large. Finally, note that the update matrix is also an $m \times m$ matrix. In order to circumvent these problems, we exploit the fact that the gradient is of low rank (rank $n$), hence one can compute a rank-$n$ factorization and use it to support fast matrix-vector product. We also observe that each row of the Jacobian matrix can be formulated as a tensor product of two vectors, therefore we can make use of fast randomized linear algebra to approximate the tensor product efficiently. As a byproduct, we have the following technical theorem:

**Theorem 1.3** (Fast Tensor Regression, informal version of Theorem D.14). *Given two $n \times m$ matrices $U$ and $V$ with $m \gg n$ and a target vector $c \in \mathbb{R}^n$. Let $J = [\text{vec}(u_1 v_1^\top)^\top, \dots, \text{vec}(u_n v_n^\top)^\top] \in \mathbb{R}^{n \times m^2}$ where $u_i$ is the $i$-th row of matrix $U$ $\forall i \in [n]$. There is a randomized algorithm that takes $\widetilde{O}(nm + n^2(\log(\kappa/\epsilon) + \log(m/\delta)) + n^\omega)$ time and outputs a vector $\widehat{x} \in \mathbb{R}^n$ such that*

$$\|JJ^\top \widehat{x} - c\|_2 \leq \epsilon\|c\|_2$$

*holds with probability at least $1 - \delta$, and $\kappa$ is the condition number of $J$.*

From a high level, the algorithm proceeds as follows: given matrices $U$ and $V$, it forms an approximation $\widetilde{J} \in \mathbb{R}^{n \times n \log(m/\delta)}$, where each row is generated by applying fast tensor sketching technique to $u_i$ and $v_i$ (Ahle et al. (2020)). Then, it uses another sketching matrix for $\widetilde{J}$ to obtain a good preconditioner $R$ for $\widetilde{J}$. Subsequently, it runs a gradient descent to solve the regression.

To understand this runtime better, we note that $nm$ term is the size of matrices $U$ and $V$, hence reading the entries from these matrices will take at least $O(nm)$ time. The algorithm then uses tensor-based sketching techniques (Ahle et al. (2020)) to squash length $m^2$ tensors to length $O(n \log(m/\epsilon\delta))$. All subsequent operations are performed on these much smaller vectors. Finally, computing the preconditioner takes $\widetilde{O}(n^\omega)$ time, and running the gradient descent takes $\widetilde{O}(n^2 \log(\kappa/\epsilon))$ time.

## 1.2 RELATED WORK

**Second-order Method in Optimization.** Though not as prevalent as first-order method in deep learning, second-order methods are one of the most popular in convex setting, such as linear programming (Vaidya (1989); Daitch & Spielman (2008); Lee et al. (2015); Cohen et al. (2019)), empirical risk minimization (Lee et al. (2019b)), cutting plane method (Jiang et al. (2020b)) and semidefinite programming (Jiang et al. (2020a)). Due to the prohibitive high cost of implementing one step of second-order method, most of these works focus on improving the *cost per iteration*.

In non-convex setting, there's a vast body of ongoing works (Martens & Grosse (2015); Botev et al. (2017); Pilanci & Wainwright (2017); Agarwal et al. (2017); Bernacchia et al. (2018); Cai et al. (2019); Zhang et al. (2019); Brand et al. (2021); Yao et al. (2021)) that try to improve the practicality of second-order method and adapt them to train deep neural networks. As shown in Cai et al. (2019), it is possible to exploit the equivalence between over-parametrized networks and neural tangent kernel to optimize an $n \times n$ matrix instead of an $m^2 \times m^2$ matrix, which is an important breakthrough in gaining speedup for second-order method. Sketching and sampling-based methods can also be used to accelerate the computation of inverses of the Hessian matrix (Pilanci & Wainwright (2017)). In spirit, our work resembles most with Cai et al. (2019) and Brand et al. (2021), in the sense that our optimization also works on an $n \times n$ Gram matrix. Our algorithm also makes use of sketching and sampling, as in Pilanci & Wainwright (2017); Brand et al. (2021).

**Over-parameterized Neural Networks.** In recent deep learning literature, understanding the geometry and convergence behavior of various optimization algorithms on over-parameterized neural networks has received a lot of attention (Li & Liang (2018); Du et al. (2019b); Allen-Zhu et al. (2019a;b); Du et al. (2019a); Song & Yang (2019); Ji & Telgarsky (2020); Zou et al. (2018); Cao & Gu (2019); Liu et al. (2020; 2021)). The seminal work of Jacot et al. (2018) initiates the study of *neural tangent kernel* (NTK), which is a powerful analytical tool in this area, since as long as the neural network is wide enough $(m \geq \Omega(n^4))$, then the optimization dynamic on a neural network is equivalent to that on a NTK.

**Sketching.** Using randomized linear algebra to reduce the dimension of the problem and speedup the algorithms for various problems has been a growing trend in machine learning community (Sarlos (2006); Clarkson & Woodruff (2013); Woodruff (2014)) due to its wide range of applications to various tasks, especially the efficient approximation of kernel matrices (Avron et al. (2014); Ahle et al. (2020); Woodruff & Zandieh (2020)). The standard "Sketch-and-Solve" (Clarkson & Woodruff (2013)) paradigm involves using sketching to reduce the dimension of the problem and then using a blackbox for the original problem to gain an edge on computational efficiency. Another line of work is to use sketching as a preconditioner (Woodruff (2014); Brand et al. (2021)) to obtain a high precision solution.

**Roadmap.** In Section 2, we give a preliminary view of the training setup we consider in this paper. In Section 2.1, we introduce the notations that will be used throughout this paper. In Section 2.2, we consider the training setup. In Section 3, we overview the techniques employed in this paper. In Section 3.2, we demonstrate various techniques to prove the convergence of our second-order method. In Section 3.1, we examine the algorithmic tools utilized in this paper to achieve subquadratic cost per iteration. In Section 4, we summarize the results in this paper and point out some future directions.

## 2 PRELIMINARIES

### 2.1 NOTATIONS

For any positive integer $n$, we use $[n]$ to denote the set $\{1, 2, \cdots, n\}$. We use $\mathbb{E}[\cdot]$ to denote expectation and $\Pr[\cdot]$ for probability. We use $\|x\|_2$ to denote the $\ell_2$ norm of a vector $x$. We use $\|A\|$ to denote the spectral norm of matrix $A$. We use $\|A\|_F$ to denote the Frobenius norm of $A$. We use $A^\top$ to denote the transpose of matrix $A$. We use $I_m$ to denote the identity matrix of size $m \times m$. For matrix $A$ or vector $x$, we use $\|A\|_0, \|x\|_0$ to denote the number of nonzero entries of $A$ and $x$ respectively. Note that $\|\cdot\|_0$ is a semi-norm since it satisfies triangle inequality. Given a real square matrix $A$, we use $\lambda_{\max}(A)$ and $\lambda_{\min}(A)$ to denote its largest and smallest eigenvalues respectively. Given a real matrix $A$, we use $\sigma_{\max}(A)$ and $\sigma_{\min}(A)$ to denote its largest and smallest singular

values respectively. We use $\mathcal{N}(\mu, \sigma^2)$ to denote the Gaussian distribution with mean $\mu$ and variance $\sigma^2$. We use $\widetilde{O}(f(n))$ to denote $O(f(n) \cdot \mathrm{poly}\log(f(n)))$. We use $\langle \cdot, \cdot \rangle$ to denote the inner product, when applying to two vectors, this denotes the standard dot product between two vectors, and when applying to two matrices, this means $\langle A, B \rangle = \mathrm{tr}[A^\top B]$ where $\mathrm{tr}[A]$ denote the trace of matrix $A$.

## 2.2 Problem Setup

Let $X \in \mathbb{R}^{m_0 \times n}$ denote the data matrix with $n$ data points and $m_0$ features. Without loss of generality, we assume $\|x_i\|_2 = 1, \forall i \in [n]$. Consider an $L$ layer neural network with one vector $a \in \mathbb{R}^{m_L}$ and $L$ matrices $W_L \in \mathbb{R}^{m_L \times m_{L-1}}, \cdots, W_2 \in \mathbb{R}^{m_2 \times m_1}$ and $W_1 \in \mathbb{R}^{m_1 \times m_0}$. We will use $W_\ell(t)$ to denote the weight matrix at layer $\ell$ at time $t$, and $\nabla W_\ell(t)$ to denote its gradient. We also use $W(t) = \{W_1(t), \ldots, W_L(t)\}$ to denote the collection of weight matrices at time $t$.

**Architecture.** We first describe our network architecture. The network consists of $L$ hidden layers, each represented by a weight matrix $W_\ell \in \mathbb{R}^{m_\ell \times m_{\ell-1}}$ for any $\ell \in [L]$. The output layer consists of a vector $a \in \mathbb{R}^{m_L}$. We define the neural network prediction function $f : \mathbb{R}^{m_0} \to \mathbb{R}$ as follows:

$$f(W, x) = a^\top \phi(W_L(\phi(\cdots \phi(W_1 x)))),$$

where $\phi : \mathbb{R} \to \mathbb{R}$ is the shifted ReLU activation function ($\sigma_b(x) = \max\{x - b, 0\}$) applied coordinate-wise to a vector.

We measure the loss via squared-loss function:

$$\mathcal{L}(W) = \frac{1}{2} \sum_{i=1}^n (y_i - f(W, x_i))^2.$$

This is also the objective function for our training.

We define the prediction function $f_t : \mathbb{R}^{m_0 \times n} \to \mathbb{R}^n$ as

$$f_t(X) = \begin{bmatrix} f(W(t), x_1) & f(W(t), x_2) & \cdots & f(W(t), x_n) \end{bmatrix}^\top.$$

**Initialization.** Our neural networks are initialized as follows:

- For each $\ell \in [L]$, the initial weight matrix $W_\ell(0) \in \mathbb{R}^{m_\ell \times m_{\ell-1}}$ is initialized such that each entry is sampled from $\mathcal{N}(0, \frac{2}{m_\ell})$.

- Each entry of $a$ is an i.i.d. sample from $\{-\frac{1}{\sqrt{m_L}}, +\frac{1}{\sqrt{m_L}}\}$ uniformly at random.

**Gradient.** In order to write gradient in an elegant way, we define some artificial variables:

$$
\begin{aligned}
g_{i,1} &= W_1 x_i, & h_{i,1} &= \phi(W_1 x_i), & &\forall i \in [n] \\
g_{i,\ell} &= W_\ell h_{i,\ell-1}, & h_{i,\ell} &= \phi(W_\ell h_{i,\ell-1}), & &\forall i \in [n], \forall \ell \in [L] \backslash \{1\} \\
D_{i,1} &= \mathrm{diag}\big(\phi'(W_1 x_i)\big), & & & &\forall i \in [n] \\
D_{i,\ell} &= \mathrm{diag}\big(\phi'(W_\ell h_{i,\ell-1})\big), & & & &\forall i \in [n], \forall \ell \in [L] \backslash \{1\}
\end{aligned} \tag{1}
$$

Using the definitions of $f$ and $h$, we have

$$f(W, x_i) = a^\top h_{i,L}, \quad \in \mathbb{R}, \quad \forall i \in [n]$$

We can compute the gradient of $\mathcal{L}$ in terms of $W_\ell \in \mathbb{R}^{m_\ell \times m_{\ell-1}}$, for all $\ell \geq 2$

$$\frac{\partial \mathcal{L}(W)}{\partial W_\ell} = \sum_{i=1}^n (f(W, x_i) - y_i) \underbrace{D_{i,\ell}}_{m_\ell \times m_\ell} \left( \prod_{k=\ell+1}^L \underbrace{W_k^\top}_{m_{k-1} \times m_k} \underbrace{D_{i,k}}_{m_k \times m_k} \right) \underbrace{a}_{m_L \times 1} \underbrace{h_{i,\ell-1}^\top}_{1 \times m_{\ell-1}} \tag{2}$$

Note that the gradient for $W_1 \in \mathbb{R}^{m_1 \times m_0}$ (recall that $m_0 = d$) is slightly different and can not be written by general form. By the chain rule, we can compute the gradient with respect to $W_1$,

$$\frac{\partial \mathcal{L}(W)}{\partial W_1} = \sum_{i=1}^n (f(W, x_i) - y_i) \underbrace{D_{i,1}}_{m_1 \times m_1} \left( \prod_{k=2}^L \underbrace{W_k^\top}_{m_{k-1} \times m_k} \underbrace{D_{i,k}}_{m_k \times m_k} \right) \underbrace{a}_{m_L \times 1} \underbrace{x_i^\top}_{1 \times m_0} \tag{3}$$

It is worth noting that the gradient matrix is of rank $n$, since it's a sum of $n$ rank-1 matrices.

**Jacobian.** For each layer $\ell \in [L]$ and time $t \in [T]$, we define the Jacobian matrix $J_{\ell,t} \in \mathbb{R}^{n \times m_\ell m_{\ell-1}}$ via the following formulation:

$$J_{\ell,t} := \left[ \mathrm{vec}(\tfrac{\partial f(W(t), x_1)}{\partial W_\ell(t)}) \quad \mathrm{vec}(\tfrac{\partial f(W(t), x_2)}{\partial W_\ell(t)}) \quad \cdots \quad \mathrm{vec}(\tfrac{\partial f(W(t), x_n)}{\partial W_\ell(t)}) \right]^\top .$$

The Gram matrix at layer $\ell$ and time $t$ is then defined as $G_{\ell,t} = J_{\ell,t} J_{\ell,t}^\top \in \mathbb{R}^{n \times n}$ whose $(i, j)$-th entry is $\langle \frac{\partial f(W(t), x_i)}{\partial W_\ell}, \frac{\partial f(W(t), x_j)}{\partial W_\ell} \rangle$.

## 3 TECHNIQUE OVERVIEW

In this section, we give an overview of the techniques employed in this paper. In Section 3.1, we showcase our algorithm and explain various techniques being used to obtain a subquadratic cost per iteration. In Section 3.2, we give an overview of the proof to show the convergence of our algorithm. To give a simpler and cleaner presentation, we assume $m_\ell = m$ for all $\ell \in [L]$.

### 3.1 SUBQUADRATIC TIME

In this section, we overview the techniques deployed in our implementation of the second-order method. Our main focus is to achieve subquadratic cost per iteration. Instead of using a Hessian matrix of size $m^2 \times m^2$, we use an $n \times n$ Gram matrix derived from the neural tangent kernel. However this would still incur a cost of $O(nm^2)$ per iteration, since each gradient is an $m \times m$ matrix and the Jacobian consists of $n$ such gradients.

We start by demonstrating our algorithm:

---

**Algorithm 1** Informal version of our algorithm.

---

1: **procedure** OURALGORITHM($f, \{x_i, y_i\}_{i \in [n]}$)             ▷ Theorem 1.1,1.2
2:      /*Initialization*/
3:      Initialize $W_\ell(0), \forall \ell \in [L]$
4:      Store $W_1(0)x_i$ in memory, $\forall i \in [n]$               ▷ Takes $O(nm^2)$ time
5:      Store $h_{i,\ell}(0) \leftarrow \phi(W_\ell(0)h_{i,\ell-1}(0)), \forall \ell \in [L], i \in [n]$ in memory    ▷ Takes $O(nLm^2)$ time
6:      **for** $t = 0 \to T$ **do**
7:          /*Forward computation*/
8:          **for** $\ell = 1 \to L$ **do**
9:              $v_{i,\ell} \leftarrow h_{i,\ell-1}, \forall i \in [n]$
10:              $h_{i,\ell} \leftarrow \phi((W_\ell(0) + \Delta W_\ell)(h_{i,\ell-1})), \forall i \in [n]$     ▷ Takes $O(n^2m) + o(nm^2)$ time
11:                                                 ▷ $h_{i,\ell}$ is sparse
12:              $D_{i,\ell} \leftarrow \mathrm{diag}(\phi'((W_\ell(0) + \Delta W_\ell)h_{i,\ell-1})), \forall i \in [n]$     ▷ Takes $O(nm)$ time
13:                                                   ▷ $D_{i,\ell}$ is sparse
14:          **end for**
15:          $f_t \leftarrow [a^\top h_{1,L}, \ldots, a^\top h_{n,L}]^\top$               ▷ Takes $O(nm)$ time
16:          /*Backward computation*/
17:          **for** $\ell = L \to 1$ **do**
18:              $u_{i,\ell} \leftarrow a^\top D_{i,L} W_L(t) \ldots D_{i,\ell+1} W_{\ell+1}(t) D_{i,\ell}$     ▷ Takes $o(nLm^2)$ time
19:              Form $\widetilde{J}_{\ell,t}$ that approximates $J_{\ell,t}$ using $\{u_{i,\ell}\}_{i=1}^n, \{v_{i,\ell}\}_{i=1}^n$
20:                              ▷ Takes $\widetilde{O}(mn)$ time, $\widetilde{J}_{\ell,t} \in \mathbb{R}^{n \times s}$ where $s = \widetilde{O}(n)$
21:              Compute $g_\ell$ that approximates $(\widetilde{J}_{\ell,t} \widetilde{J}_{\ell,t}^\top)^{-1} c$            ▷ Takes $\widetilde{O}(nm)$ time
22:              Form $J_{\ell,t}^\top g_\ell$ via low rank factorization $\sum_{i=1}^n g_{\ell,i} u_{i,\ell} v_{i,\ell}^\top$
23:              Implicitly update $\Delta W_\ell \leftarrow \Delta W_\ell + \sum_{i=1}^n g_{\ell,i} u_{i,\ell} v_{i,\ell}^\top$ and store it in memory
24:          **end for**
25:      **end for**
26: **end procedure**

---

**Step 1: Invert Gram by solving regression.** Recall the update rule of generic algorithm is given by

$$W_\ell(t+1) \leftarrow W_\ell(t) - J_{\ell,t}^\top (J_{\ell,t} J_{\ell,t}^\top) c,$$

where $c$ is $f_t - y$ after proper scaling. Naively forming the Gram matrix $J_{\ell,t} J_{\ell,t}^\top$ will take $O(n^2 m^2)$ time and inverting it will take $O(n^\omega)$ time. To avoid the quadratic cost at this step, we instead solve a regression, or a linear system since the Gram matrix has full rank: find the vector $g_{\ell,t} \in \mathbb{R}^n$ such that

$$\|J_{\ell,t} J_{\ell,t}^\top g_{\ell,t} - c\|_2^2$$

is minimized. This enables us to utilize the power of sketching to solve the regression efficiently.

**Step 2: Solve Gram regression via preconditioning.** In order to solve the above regression, we adapt the idea of obtaining a good preconditioner via sketching then apply iterative method to solve it (Brand et al. (2021)). Roughly speaking, we first use a random matrix $S \in \mathbb{R}^{s \times m^2}$ that has the *subspace embedding property* (Sarlos (2006)) to reduce the number of rows of $J^\top$, then we run a QR decomposition on matrix $SJ^\top$. This gives us a matrix $R$ such that $SJ^\top R$ has orthonormal columns. We then use gradient descent to optimize the objective $\|JJ^\top Rz_t - y\|_2^2$. Since $S$ is a subspace embedding for $J^\top$, we can make sure that the condition number of the matrix $J^\top R$ is small ($O(1)$), hence the gradient descent converges after $\log(\kappa/\epsilon)$ iterations, where $\kappa$ is the condition number of $J$. However, in order to implement the gradient descent, we still need to multiply an $m^2 \times n$ matrix with a length $n$ vector, in the worst case this will incur a time of $O(nm^2)$. In order to bypass this barrier, we need to exploit extra structural properties of the Jocobian, which will be demonstrated in the following steps.

**Step 3: Low rank structure of the gradient.** Instead of studying Jacobian directly, we first try to understand the *low rank structure* of the gradient. Consider $\frac{\partial f(W, x_i)}{\partial W_\ell} \in \mathbb{R}^{m \times m}$, it can be written as (for simplicity, we use $h_{i,0}$ to denote $x_i$):

$$\frac{\partial f(W, x_i)}{\partial W_\ell} = \underbrace{h_{i,\ell-1}}_{v_i \in \mathbb{R}^{m \times 1}} \underbrace{a^\top D_{i,L} W_L \dots D_{i,\ell+1} W_{i,\ell+1} D_{i,\ell}}_{u_i^\top \in \mathbb{R}^{1 \times m}}.$$

This means the gradient is essentially an outer product of two vectors, and hence has rank one. This has several interesting consequences: for over-parametrized networks, the gradient is merely of rank $n$ instead of $m$. When using first-order method such as gradient descent or stochastic gradient descent, the weight is updated via a low rank matrix. To some extent, this explains why the weight does not move too far from initialization in over-parametrized networks when using first-order method to train. Also, as we will illustrate below, this enables the efficient approximation of Jacobian matrices and maintenance of the change.

**Step 4: Fast approximation of the Jacobian matrix.** We now turn our attention to design a fast approximation algorithm to the Jacobian matrix. Recall that Jacobian matrix $J_{\ell,t} \in \mathbb{R}^{n \times m^2}$ is an $n \times m^2$ matrix, therefore writing down the matrix will take $O(nm^2)$ time. However, it is worth noticing that each row of $J_{\ell,t}$ is $\text{vec}(u_i v_i^\top)^\top, \forall i \in [n]$, or equivalently, $u_i \circ v_i$ where $\circ$ denotes the tensor product between two vectors. Suppose we are given the collection of $\{u_1, \dots, u_n\} \in (\mathbb{R}^m)^n$ and $\{v_1, \dots, v_n\} \in (\mathbb{R}^m)^n$, then we can compute the tensor product $u_i \circ v_i$ via tensor-based sketching techniques, such as TensorSketch (Avron et al. (2014); Diao et al. (2017; 2019)) or TensorSRHT (Ahle et al. (2020); Woodruff & Zandieh (2020)) in time nearly linear in $m$ and the targeted sketching dimension $s$, in contrast to the naive $O(m^2)$ time. Since it suffices to preserve the length of all vectors in the column space of $J_{\ell,t}^\top$, the target dimension $s$ can be chosen as $O(\epsilon^{-2} n \cdot \text{poly}(\log(m/\epsilon\delta)))$. Use $\widetilde{J}_{\ell,t} \in \mathbb{R}^{n \times s}$ to denote this approximation of $J_{\ell,t}$, we perform the preconditioned gradient descent we described above on this smaller matrix. This enables to lower the overall cost of the regression step to be subquadratic in $m$.

**Step 5: Efficient update via low rank factorization.** The low rank structure of the gradient can further be utilized to represent the change on weight matrices $\Delta W$ in a way such that any matrix-vector product involving $\Delta W$ can be performed fast. Let $g_\ell \in \mathbb{R}^n$ denote the solution to the regression problem posed in Step 1. Note that by the update rule of our method, we shall use $J_{\ell,t}^\top g_\ell \in \mathbb{R}^{m \times m}$ to update the weight matrix, but writing down the matrix will already take $O(m^2)$ time. Therefore,

it is instructive to find a succinct representation for the update. The key observation is that each column of $J_{\ell,t}^\top$ is a tensor product of two vectors: $u_i \circ v_i$ or equivalently, $u_i v_i^\top$. The update matrix can be rewritten as $\sum_{i=1}^n g_{\ell,i} u_i v_i^\top$, and we can use this representation for the update on the weight, instead of adding it directly. Let

$$U_\ell := \begin{bmatrix} | & | & \cdots & | \\ g_{\ell,1}u_1 & g_{\ell,2}u_2 & \cdots & g_{\ell,n}u_n \\ | & | & \cdots & | \end{bmatrix} \in \mathbb{R}^{m\times n}, V_\ell := \begin{bmatrix} | & | & \cdots & | \\ v_1 & v_2 & \cdots & v_n \\ | & | & \cdots & | \end{bmatrix} \in \mathbb{R}^{m\times n},$$

then the update can be represented as $U_\ell V_\ell^\top$. Consider multiplying a vector $y \in \mathbb{R}^m$ with this representation, we first multiply $y$ with $V_\ell^\top \in \mathbb{R}^{n\times m}$, which takes $O(mn)$ time. Then we multiply $V_\ell^\top y \in \mathbb{R}^n$ with $U_\ell \in \mathbb{R}^{m\times n}$ which takes $O(mn)$ time. This drastically reduces the cost of multiplying the weight matrix with a vector from $O(m^2)$ to $O(mn)$.

Inspired by this idea, it is tempting to store all intermediate low rank representations across all iterations and use them to facilitate matrix-vector product, which incurs a runtime of $O(Tmn)$. This is fine when $T$ is relatively small, however, if one looks for a high precision solution which requires a large number of iterations, then $T$ might be too large and $O(Tmn)$ might be in the order of $O(m^2)$. To circumvent this problem, we design the data structure so that it will exactly compute the $m \times m$ change matrix and update the weight and clean up the cumulative changes. This can be viewed as a "restart" of the data structure. To choose the correct number of updates before restarting, we utilize the *dual exponent of matrix multiplication*, $\alpha$ (Gall & Urrutia, 2018), which means it takes $O(m^2)$ time to multiply an $m \times m$ by an $m \times m^\alpha$ matrix. Hence, we restart the data structure after around $m^\alpha/n$ updates. Therefore, we achieve an amortized $o(m^2)$ time, which is invariant even though the number of iterations $T$ grows larger and larger.

### 3.2 CONVERGENCE ANALYSIS

In this section, we demonstrate the strategy to prove that our second-order method achieves a linear convergence rate on the training loss.

**Step 1: Initialization.** Let $W(0)$ be the random initialization. We first show that for any data point $x_i$, we have $f(W(0), x_i) = O(1)$. The analysis draws inspiration from Allen-Zhu et al. (2019a). The general idea is, given a fixed unit length vector $x$, multiplying it with a random Gaussian matrix $W$ will make sure that $\|Wx\|_2^2 \approx 2$. Since $W$ is a random Gaussian matrix, applying shifted ReLU activation gives a random vector with a truncated Gaussian distribution conditioned on a binomial random variable indicating which neurons are activated. We will end up with $\|\phi(Wx)\|_2 \approx 1$ as well as $\phi(Wx)$ being sparse. Inductively applying this idea to each layer and carefully controlling the error occurring at each layer, we can show that with good probability, $f(W(0), x_i)$ is a constant.

We also bound the spectral norm of $\frac{\partial f(W(0), x_i)}{\partial W_\ell}$ by $\widetilde{O}(\sqrt{L/m})$. One of the key part of this matrix is the consecutive product $D_L W_L \ldots D_{\ell+1} W_{\ell+1} D_\ell$. By studying the distribution of its product with a fixed vector, one can show that the spectral norm of this consecutive product is bounded by $O(\sqrt{L})$. Finally, we make use the fact that each entry of $a$ is a Rademacher random variable scaled by $1/\sqrt{m}$, hence the norm of $a^\top D_L W_L \ldots D_{\ell+1} W_{\ell+1} D_\ell$ is bounded by $\widetilde{O}(\sqrt{L/m})$ with good probability.

Furthermore, we show that the Gram matrix for the multiple-layer over-parametrized neural network, which is defined as $J_{\ell,0} J_{\ell,0}^\top$, has a nontrivial minimum eigenvalue after the initialization. In particular, we adapt the neural tangent kernel (NTK) for multiple-layer neural networks defined by Du et al. (2019a) into our setting by analyzing the corresponding Gaussian process with shifted ReLU activation function. Then, we can prove that with high probability, the least eigenvalue of the initial Gram matrix is lower bounded by the least eigenvalue of the neural tangent kernel matrix.

**Step 2: Small perturbation.** The next step is to show that if all weight matrices undergo a small perturbation from initialization (in terms of spectral norm), then the corresponding Jacobian matrix has not changed too much. As long as the perturbation is small enough, it is possible to show that the change of the $h$ vector (in terms of $\ell_2$ norm) and the consecutive product (in terms of spectral norm) is also small. Finally, using the fact that $a$ is a Rademacher vector with scaling $1/\sqrt{m}$, we can show that the change

$$\frac{\partial f(W(0) + \Delta W, x_i)}{\partial(W_\ell + \Delta W_\ell)} - \frac{\partial f(W(0), x_i)}{\partial W_\ell}$$

has its spectral norm being bounded by $\widetilde{O}(\sqrt{L/m})$ for any layer $\ell \in [L]$ and input data $i \in [n]$. Consequently, the Frobenious norm of the Jacobian matrix is bounded by $\widetilde{O}(\sqrt{nL/m})$.

**Step 3: Connect everything via a double induction.** Put things together, we use a double induction argument, where we assume the perturbation of weight matrix is small and the gap between $f_t$ and $y$ is at most $1/2$ of the gap between $f_{t-1}$ and $y$. By carefully bounding various terms and exploiting the fact the Jacobian matrix *always* has a relative small spectral norm ($\widetilde{O}(\sqrt{nL/m})$), we first show that the weights are not moving too far from the initialization, then use this fact to derive a final convergence bound for $\|f_t - y\|_2$.

## 4 DISCUSSION AND FUTURE DIRECTIONS

In this work, we propose and analyze a second-order method to train multi-layer over-parametrized neural networks. Our algorithm achieves a linear convergence rate in terms of training loss, and achieves a subquadratic ($o(m^2)$) cost per training iteration. From an analytical perspective, we greatly extend the analysis of (Allen-Zhu et al. (2019a)) to second-order method, coupled with the usage of the equivalence between multi-layer over-parametrized networks and neural tangent kernels (Du et al. (2019a)). From an algorithmic perspective, we achieve a subquadratic cost per iteration, which is a significant improvement from $O(m^2 n)$ time per iteration due to the prohibitively large network width $m$. Our algorithm combines various techniques, such as training with the Gram matrix, solve the Gram regression via sketching-based preconditioning, fast tensor computation and dimensionality reduction and low rank decomposition of weight updates. Our algorithm is especially valuable when one requires a high precision solution on training loss, and hence the number of iterations is large.

One of the interesting questions from our work is: is it possible to obtain an algorithm that has a *nearly linear* cost per iteration on $m$ as in the case of training one-hidden layer over-parametrized networks (Brand et al. (2021))? In particular, can this runtime be achieved under the current best width of multi-layer over-parametrized networks ($m \geq n^8$)? We note that the major limitation in our method is the *sparsity* of the change of the diagonal matrices ($\Delta D$) is directly related to the magnitude of the change of weights ($\|\Delta W\|$). In our analysis of convergence, we go through a careful double induction argument, which in fact imposes on a lower bound on $\|\Delta W\|$. It seems to us that, in order to achieve a nearly linear runtime, one has to adapt a different analytical framework or approach the problem from a different perspective.

A related question is, how can we maintain the changes of weight more efficiently? In our work, we achieve speedup in the neural network training process by observing that the change of the weights are small in each iteration. Similar phenomenon also appears in some classical optimization problem (e.g., solving linear program (Cohen et al., 2019; Jiang et al., 2021) and solving semidefinite program (Jiang et al., 2020a)) and they achieve further speedup by using lazy update and amortization techniques to compute the weight changes, or using more complicated data structure to maintain the *changes* of the weight changes. Can we adapt their techniques to neural network training? An orthogonal direction to maintain the change is to design an initialization setup such that while we still have enough randomness to obtain provable guarantees, the matrix-vector product with the initial weight matrix can be performed faster than $O(m^2)$ by sparsifying the Gaussian matrix as in Dereziński et al. (2021) or imposing extra structural assumption such as using circulant Gaussian (Rauhut et al., 2012; Nelson & Nguyẫn, 2013; Krahmer et al., 2014).

Another question concerns activation functions. In this paper, we consider the shifted ReLU activation and design our algorithm and analysis around its properties. Is it possible to generalize our algorithm and analysis to various other activation functions, such as sigmoid, tanh or leaky ReLU? If one chooses a smooth activation, can we get a better result in terms of convergence rate? Can we leverage this structure to design faster algorithms?

Finally, the network architecture considered in this paper is the standard feedforward network. Is it possible to extend our analysis and algorithm to other architectures, such as recurrent neural networks (RNN)? For RNN, the weight matrices for each layer are the same, hence it is trickier to analyze the training dynamics on such networks. Though the convergence of first-order method on over-parametrized multi-layer RNN has been established, it is unclear whether such analysis can be extended to second-order method.

**Ethics Statement.** This is a theory paper that proposed efficient algorithm to train multi-layer over-parametrized neural networks, hence it does not have direct ethics implications.

**Reproducibility Statement.** All results in this paper can be directly verified and reproduced via reading into the proofs. For complete algorithm description and its runtime analysis, see Section B. For specific data structure used in the algorithm, see Section C. For the result regarding the fast tensor regression, see Section D. For convergence analysis of the algorithm, see Section E, F and G.

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

**Roadmap.** In Section A, we remind readers with the notations and some probability tools. In Section B, we illustrate the complete version of our algorithm and give a runtime analysis of it. In Section C, we design a simple low rank maintenance data structure and show how to use it to efficiently implement matrix-vector product. In Section D, we introduce an efficient regression solver handling our Jacobian and Gram regression. In Section E, we study the spectral property of the Gram matrix at each layer and connects it with multi-layer neural tangent kernels. In Section F, we analyze the convergence of our algorithm by using some heavy machinery such as structural analysis of the gradient and a careful double induction. In Section G, we give a detailed proof of one technical lemma.

## A PRELIMINARIES AND PROBABILITY TOOLS

In this section, we introduce notations that will be used throughout the rest of the paper and several useful probability tools that will be heavily exploited in the later proofs.

**Notations.** For any positive integer $n$, we use $[n]$ to denote the set $\{1, 2, \cdots, n\}$. We use $\mathbb{E}[\cdot]$ to denote expectation and $\Pr[\cdot]$ for probability. We use $\|x\|_2$ to denote the $\ell_2$ norm of a vector $x$. We use $\|A\|$ to denote the spectral norm of matrix $A$. We use $\|A\|_F$ to denote the Frobenius norm of $A$. We use $A^\top$ to denote the transpose of matrix $A$. We use $I_m$ to denote the identity matrix of size $m \times m$. For matrix $A$ or vector $x$, we use $\|A\|_0, \|x\|_0$ to denote the number of nonzero entries of $A$ and $x$ respectively. Note that $\|\cdot\|_0$ is a semi-norm since it satisfies triangle inequality. Given a real square matrix $A$, we use $\lambda_{\max}(A)$ and $\lambda_{\min}(A)$ to denote its largest and smallest eigenvalues respectively. Given a real matrix $A$, we use $\sigma_{\max}(A)$ and $\sigma_{\min}(A)$ to denote its largest and smallest singular values respectively. We use $\mathcal{N}(\mu, \sigma^2)$ to denote the Gaussian distribution with mean $\mu$ and variance $\sigma^2$. We use $\widetilde{O}(f(n))$ to denote $O(f(n) \cdot \operatorname{poly}\log(f(n)))$. We use $\langle \cdot, \cdot \rangle$ to denote the inner product, when applying to two vectors, this denotes the standard dot product between two vectors, and when applying to two matrices, this means $\langle A, B \rangle = \operatorname{tr}[A^\top B]$ where $\operatorname{tr}[A]$ denote the trace of matrix $A$.

**Lemma A.1** (Chernoff bound Chernoff (1952)). *Let $X = \sum_{i=1}^n X_i$, where $X_i = 1$ with probability $p_i$ and $X_i = 0$ with probability $1 - p_i$, and all $X_i$ are independent. Let $\mu = \mathbb{E}[X] = \sum_{i=1}^n p_i$. Then*
*1. $\Pr[X \geq (1+\delta)\mu] \leq \exp(-\delta^2\mu/3), \forall \delta > 0$ ;*
*2. $\Pr[X \leq (1-\delta)\mu] \leq \exp(-\delta^2\mu/2), \forall 0 < \delta < 1$.*

**Lemma A.2** (Hoeffding bound Hoeffding (1963)). *Let $X_1, \cdots, X_n$ denote $n$ independent bounded variables in $[a_i, b_i]$. Let $X = \sum_{i=1}^n X_i$, then we have*

$$\Pr[|X - \mathbb{E}[X]| \geq t] \leq 2\exp\left(-\frac{2t^2}{\sum_{i=1}^n (b_i - a_i)^2}\right).$$

**Lemma A.3** (Bernstein inequality Bernstein (1924)). *Let $X_1, \cdots, X_n$ be independent zero-mean random variables. Suppose that $|X_i| \leq M$ almost surely, for all $i$. Then, for all positive $t$,*

$$\Pr\left[\sum_{i=1}^n X_i > t\right] \leq \exp\left(-\frac{t^2/2}{\sum_{j=1}^n \mathbb{E}[X_j^2] + Mt/3}\right).$$

**Lemma A.4** (Anti-concentration of Gaussian distribution). *Let $X \sim \mathcal{N}(0, \sigma^2)$, then*

$$\Pr[|X| \leq t] \in (\frac{2}{3}\frac{t}{\sigma}, \frac{4}{5}\frac{t}{\sigma}).$$

**Lemma A.5** (Concentration of subgaussian random variables). *Let $a \in \mathbb{R}^n$ be a vector where each entry of $a$ is sampled from a subgaussian distribution of parameter $\sigma^2$, then for any vector $x \in \mathbb{R}^n$,*

$$\Pr[|\langle a, x \rangle| \geq t \cdot \|x\|_2] \leq 2\exp(-\frac{t^2}{2\sigma^2}).$$

**Lemma A.6** (Small ball probability). *Let $a \in \mathbb{R}^n$ be a vector such that $|a_i| \geq \delta$ for all $i \in [n]$, $x_1, \ldots, x_n$ are $n$ i.i.d. Rademacher random variables. Then, there exist absolute constants $C_1, C_2$ such that for any $t > 0$,*

$$\Pr[|\langle a, x \rangle| \leq t] \leq \min\left\{\frac{C_1 t}{\|a\|_2}, \frac{C_2 t}{\delta\sqrt{n}}\right\}.$$

# B  COMPLETE ALGORITHM AND ITS RUNTIME ANALYSIS

In this section, we first present our complete algorithm, then give a runtime analysis of it.

---

**Algorithm 2** Complete version of our algorithm.

---

1: **procedure** COMPLETEALGORITHM($X \in \mathbb{R}^{d \times n}, y \in \mathbb{R}^n$)                    ▷ Theorem B.1
2:     /*Initialization*/
3:     Initialize $W_\ell(0), \forall \ell \in [L]$
4:     Store $W_1(0)x_i$ in memory, $\forall i \in [n]$                    ▷ Takes $O(nm^2)$ time
5:     LOWRANKMAINTENANCE LMR                    ▷ Algorithm 3
6:     LMR.INIT($\{W_1(0)\ldots, W_L(0)\}$)
7:     **for** $t = 0 \to T$ **do**
8:         /*Forward computation*/
9:         **for** $\ell = 1 \to L$ **do**
10:             $v_{i,\ell} \leftarrow h_{i,\ell-1}, \forall i \in [n]$
11:             $g_{i,\ell} \leftarrow$ LMR.QUERY($\ell, h_{i,\ell-1}$)                    ▷ Takes $o(nm^2)$ time
12:             $h_{i,\ell} \leftarrow \phi(g_{i,\ell}), \forall i \in [n]$
13:                                 ▷ $h_{i,\ell}$ is sparse
14:             $D_{i,\ell} \leftarrow \mathrm{diag}(\phi'(g_{i,\ell})), \forall i \in [n]$                    ▷ Takes $O(nm)$ time
15:                                 ▷ $D_{i,\ell}$ is sparse
16:         **end for**
17:         $f_t \leftarrow [a^\top h_{1,L}, \ldots, a^\top h_{n,L}]^\top$                    ▷ Takes $O(nm)$ time
18:         /*Backward computation*/
19:         **for** $\ell = L \to 1$ **do**
20:             $u_{i,\ell} \leftarrow a^\top D_{i,L}W_L(t)\ldots D_{i,\ell+1}W_{\ell+1}(t)D_{i,\ell}, \forall i \in [n]$         ▷ Takes $o(nLm^2)$ time
21:             $g_\ell \leftarrow$ FASTTENSORREGRESSION($\{u_{i,\ell}\}_{i=1}^n, \{v_{i,\ell}\}_{i=1}^n, c$) with precision $\sqrt{\lambda/n}$
22:                                 ▷ Algorithm 6
23:             LMR.UPDATE($\{g_{\ell,i}u_{i,\ell}\}_{i=1}^n, \{v_{i,\ell}\}_{i=1}^n$)
24:         **end for**
25:     **end for**
26: **end procedure**

---

**Theorem B.1** (Formal version of Theorem 1.2)**.** *Let $X \in \mathbb{R}^{d \times n}$ and $y \in \mathbb{R}^n$, and let $k$ denote the sparsity of $D_{i,\ell}$ and $s$ denote the sparsity of $\Delta D_{i,\ell}$, $\forall \ell \in [L], i \in [n]$. Let $m$ denote the width of neural network, $L$ denote the number of layers and $\alpha$ denote the dual matrix multiplication exponent (Def. C.1),then the running time of Algorithm 2 is*

$$O(\mathcal{T}_{\mathrm{init}} + T \cdot \mathcal{T}_{\mathrm{iter}}),$$

*where*

$$\mathcal{T}_{\mathrm{init}} = O(m^2(n + L)),$$
$$\mathcal{T}_{\mathrm{iter}} = \widetilde{O}((m^{1+\alpha} + m(s + k))L^2 + m^{2-\alpha}nL).$$

*Therefore, the* cost per iteration *of Algorithm 2 is*

$$\widetilde{O}((m^{1+\alpha} + m(s + k))L^2 + m^{2-\alpha}nL).$$

*Proof.* We analyze $\mathcal{T}_{\mathrm{init}}$ and $\mathcal{T}_{\mathrm{iter}}$ separately.

**Initialization time.** We will first initialize $(L - 1)$ $m \times m$ matrices and one $m \times d$ matrix, which takes $O(m^2 L)$ time. Compute $W_1(0)x_i$ for all $i \in [n]$ takes $O(m^2 n)$ time. Finally, initialize the data structure takes $O(m^2 L)$ time. Hence, $\mathcal{T}_{\mathrm{init}} = O(m^2(n + L))$.

**Cost per iteration.** For each iteration, we perform one forward computation from layer 1 to $L$, then backpropagate from layer $L$ to 1.

- **Forward computation:** In forward computation, we first compute $g_{i,\ell} \in \mathbb{R}^m$, which is equivalent to form $v_{i,\ell+1}$, hence by Lemma C.5, it takes $O(s + k + m^\alpha)$ time. Compute $h_{i,\ell}$ and $D_{i,\ell}$ takes $O(m)$ time. These computations would be performed for all $L$ layers, hence the overall runtime of forward computation is $O((s + k + m^\alpha)L)$ time.

- **Backpropagation:** In backpropagation, we first compute $u_{i,\ell} \in \mathbb{R}^m$, which takes $O(mL(s + k + m^\alpha))$ time. Then, we call Algorithm 6 to solve the Gram regression problem, which due to Theorem D.14 takes $\widetilde{O}(mn + n^\omega)$ time. Note that even we want a high probability version of the solver with $e^{-\log^2 nL}$ failure probability so that we can union bound over all layers and all iterations, we only pay extra $\log^2 nL$ term in running time, which is absorbed by the $\widetilde{O}(\cdot)$ notation. Finally, the update takes $O(m^{2-\alpha}n)$ time by Lemma C.2. Sum over all $L$ layers, we get an overall running time of $\widetilde{O}((m^{1+\alpha} + m(s + k))L^2 + m^{2-\alpha}nL)$ time.

This concludes the proof of our Theorem. $\qquad\square$

**Corollary B.2.** *Suppose the network width $m$ is chosen as in F.25 and the shift parameter $b$ is chosen as in F.6, then the cost per iteration of Algorithm 2 is*

$$\widetilde{O}(m^{1.8}L^2 + m^{1.69}nL).$$

**Remark B.3.** *As long as the neural network is wide enough, as in F.25 and we choose the shifted threshold properly, as in F.6, then we can make sure that both sparsity parameters $k$ and $s$ to be $o(m)$, and we achieve subquadratic cost per iteration.*

## C  LOW RANK MAINTENANCE AND EFFICIENT COMPUTATION OF THE CHANGE

In this seciton, we design a data structure to maintain the low rank representation of change of weights, and then we show how to efficiently compute the low rank representation using this data structure.

### C.1  LOW RANK MAINTENANCE

In this short section, we design a simple data structure to maintain the low rank representation of change of weights, and show that the matrix-vector product can be implemented efficiently.

Before moving, we define some notions related to rectangular matrix multiplication.

**Definition C.1** (Williams (2012); Gall & Urrutia (2018))**.** *Let $\omega$ be the matrix multiplication exponent such that it takes $n^{\omega+o(1)}$ time to multiply two $n \times n$ matrices.*

*Let $\alpha$ be the dual exponent of the matrix multiplication which is the supremum among all $a \geq 0$ such that it takes $n^{2+o(1)}$ time to multiply an $n \times n$ by $n \times n^a$ matrix.*

*Additionally, we define the function $\omega(\cdot)$ where $\omega(b)$ denotes the exponent of multiplying an $n \times n$ matrix by an $n \times n^b$ matrix. Hence, we have $\omega(1) = \omega$ and $\omega(\alpha) = 2$.*

The overall idea of our low rank maintenance data structure is as follows: we keep accumulating the low rank change, when the rank of the change reaches a certain threshold ($m^\alpha$), then we restart the data structure and update the weight matrix.

---

**Algorithm 3** Low rank maintenance data structure

---

1: **data structure** LOWRANKMAINTENANCE $\quad\quad\quad\quad\quad\quad\quad\quad\quad\quad\quad\quad$ ▷ Lemma C.2
2: $\quad$ **members**
3: $\quad\quad$ $r_\ell, \forall \ell \in [L]$ $\quad\quad\quad\quad\quad\quad\quad\quad$ ▷ $r_\ell$ denotes the accumulated rank of the change
4: $\quad\quad$ $W_\ell, \forall \ell \in [L]$ $\quad\quad\quad\quad\quad\quad\quad\quad\quad\quad\quad\quad\quad$ ▷ $\{W_\ell\}_{\ell=1}^L \in (\mathbb{R}^{m \times m})^L$
5: $\quad\quad$ $\Delta W_\ell, \forall \ell \in [L]$ $\quad\quad\quad\quad\quad\quad\quad\quad\quad\quad\quad\quad$ ▷ $\{\Delta W_\ell\}_{\ell=1}^L \in (\mathbb{R}^{m \times m})^L$
6: $\quad$ **end members**
7:
8: $\quad$ **procedures**
9: $\quad\quad$ INIT($\{W_1(0), \ldots W_L(0)\}$) $\quad\quad\quad\quad\quad\quad$ ▷ Initialize the data structure
10: $\quad\quad$ UPDATE($U_\ell, V_\ell$) $\quad\quad\quad\quad\quad\quad\quad\quad$ ▷ Update the low rank representation
11: $\quad\quad$ QUERY($\ell, y$) $\quad\quad\quad\quad$ ▷ Compute the matrix-vector product between $\Delta W_\ell$ and $y$
12: $\quad$ **end procedures**
13: **end data structure**

---

**Algorithm 4** Procedures of LRM data structure

---

1: **procedure** INIT($\{W_1(0), \ldots, W_L(0)\}$) $\quad\quad\quad\quad\quad\quad\quad\quad$ ▷ Lemma C.2
2: $\quad$ $W_\ell \leftarrow W_\ell(0)$
3: $\quad$ $\Delta W_\ell \leftarrow 0, \forall \ell \in [L]$
4: $\quad$ $r_\ell \leftarrow 0, \forall \ell \in [L]$
5: **end procedure**
6:
7: **procedure** UPDATE($U_\ell \in \mathbb{R}^{m \times n}, V_\ell \in \mathbb{R}^{m \times n}$) $\quad\quad\quad\quad\quad$ ▷ Lemma C.2
8: $\quad$ $\Delta W_\ell \leftarrow \Delta W_\ell + U_\ell V_\ell^\top$ without forming the product and sum the two matrices
9: $\quad$ $r_\ell \leftarrow r_\ell + n$
10: $\quad$ **if** $r_\ell = m^a$ where $a = \omega(2)$ **then**
11: $\quad\quad$ $W_\ell \leftarrow W_\ell + \Delta W_\ell$ $\quad\quad\quad\quad\quad\quad\quad\quad\quad\quad$ ▷ Takes $O(m^2)$ time
12: $\quad\quad$ $r_\ell \leftarrow 0$
13: $\quad\quad$ $\Delta W_\ell \leftarrow 0$
14: $\quad$ **end if**
15: **end procedure**
16:
17: **procedure** QUERY($\ell \in [L], y \in \mathbb{R}^m$) $\quad\quad\quad\quad\quad\quad\quad\quad$ ▷ Lemma C.2
18: $\quad$ $z \leftarrow W_\ell \cdot y + \Delta W_\ell \cdot y$ $\quad\quad\quad\quad$ ▷ Takes $O(\text{nnz}(y) \cdot m + m r_\ell)$ time
19: $\quad$ **return** $z$
20: **end procedure**

---

**Lemma C.2.** *There exists a deterministic data structure (Algorithm 3) such that maintains $\Delta W_1, \ldots, \Delta W_L$ such that*

- *The procedure* INIT *(Algorithm 4) takes $O(m^2 L)$ time.*

- *The procedure* UPDATE *(Algorithm 4) takes $O(nm^{2-\alpha})$ amortized time, where $\alpha = \omega(2)$*

- *The procedure* QUERY *(Algorithm 4) takes $O(m \cdot (\text{nnz}(y) + r_\ell))$ time, where $r_\ell$ is the rank of $\Delta W_\ell$ when* QUERY *is called.*

*Proof.* The runtime for INIT is obvious, for QUERY, notice that we are multiplying vector $y$ with a (possibly) dense matrix $W_\ell \in \mathbb{R}^{m \times m}$, which takes $O(\text{nnz}(y) \cdot m)$ time, and an accumulated low rank matrix $\Delta W_\ell$ with rank $r_\ell$. By using the low rank decomposition $\Delta W_\ell = UV^\top$ with $U, V \in \mathbb{R}^{m \times r_\ell}$, the time to multiply $y$ with $\Delta W$ is $O(m r_\ell)$. Combine them together, we get a running time of $O(m \cdot (\text{nnz}(y) + r_\ell))$.

It remains to analyze the amortized cost of UPDATE. Note that if $r_\ell < m^a$, then we just pay $O(1)$ time to update corresponding variables in the data structure. If $r_\ell = m^a$, then we will explicitly form the $m \times m$ matrix $\Delta W_\ell$. To form it, notice we have accumulated $r_\ell / n$ different sums of

rank-$n$ decompositions, which can be represented as

$$U = [U_\ell(1), U_\ell(2), \ldots, U_\ell(r_\ell/n)] \in \mathbb{R}^{m \times r_\ell}, V = [V_\ell(1), V_\ell(2), \ldots, V_\ell(r_\ell/n)] \in \mathbb{R}^{m \times r_\ell},$$

and $\Delta W_\ell = UV^\top$, which takes $O(m^2)$ time to compute since $r_\ell = m^a$ and $a = \omega(2)$. Finally, note that this update of $W_\ell$ only happens once per $r_\ell/n$ number of calls to UPDATE, therefore we can charge each step by $O(\frac{m^2}{r_\ell/n}) = O(m^{2-a}n) = O(m^{2-\alpha}n)$, arrives at our final amortized running time. $\qquad \square$

**Remark C.3.** *Currently, the dual matrix multiplication exponent $\alpha \approx 0.31$ (Gall & Urrutia, 2018), hence the amortized time for UPDATE is $O(nm^{1.69})$. If $m \geq n^{10/3}$, then we achieve an update time of $o(m^2)$. Similarly, the time for QUERY is $O(m \cdot (\mathrm{nnz}(y) + r_\ell)) = O(m \cdot \mathrm{nnz}(y) + m^{1+\alpha}) = O(m \cdot \mathrm{nnz}(y) + m^{1.31})$, as long as $\mathrm{nnz}(y) = o(m)$, then its running time is also $o(m^2)$. In our application of training neural networks, we will make sure that the inputted vector $y$ is sparse.*

## C.2 EFFICIENTLY COMPUTE $u_{i,\ell}(t)$ AND $v_{i,\ell}(t)$

In this section, we show that how to compute the vectors $u_{i,\ell}, v_{i,\ell} \in \mathbb{R}^m$ using the low rank structure of the change of weights combined. Recall the definition of these vectors:

$$u_{i,\ell}(t)^\top = a^\top D_{i,L}(t) W_L(t) \ldots D_{i,\ell+1}(t) W_{\ell+1}(t) D_{i,\ell}(t) \in \mathbb{R}^{1 \times m},$$
$$v_{i,\ell}(t) = h_{i,\ell-1}(t) \in \mathbb{R}^m.$$

Before proceeding, we list the assumptions we will be using:

- For any $\ell \in [L]$, $D_{i,\ell}(t)$ is $s_D$-sparse, where $s_D := k + s$, $k$ is the sparsity of $D_{i,\ell}(0)$ and $s$ is the sparsity of $D_{i,\ell}(t) - D_{i,\ell}(0)$.

- For any $\ell \in [L]$, the change of the weight matrix $W_\ell$, $\Delta W_\ell(t) := W_\ell(t) - W_\ell(0)$, is of low-rank. That is, $\Delta W_\ell(t) = \sum_{j=1}^{r_t} y_{\ell,j} z_{\ell,j}^\top$.

- For any $i \in [n]$, $W_1(0)x_i$ is pre-computed.

We first note that as a direct consequence of $D_{i,\ell}(0)$ is $k$-sparse, $h_{i,\ell}(0)$ is $k$-sparse as well. Similarly, $h_{i,\ell}(t) - h_{i,\ell}(0)$ has sparsity $s$. Hence $h_{i,\ell}(t)$ has sparsity bounded by $s_D$.

**Compute $u_{i,\ell}(t)$.** Compute $u_{i,\ell}(t)$ is equivalent to compute the following vector:

$$D_{i,\ell}(t)(W_{\ell+1}(0) + \Delta W_{\ell+1}(t))^\top D_{i,\ell+1}(t) \cdots (W_L(0) + \Delta W_L(t))^\top D_{i,L}(t)a.$$

First, we know that $D_{i,L}(t)a \in \mathbb{R}^m$ is an $s_D$-sparse vector, and it takes $O(s_D)$ time. The next matrix is $(W_L(0) + \Delta W_L(t))^\top$, which gives two terms: $W_L(0)^\top (D_{i,L}(t)a)$ and $\Delta W_L(t)^\top (D_{i,L}(t)a)$. For the first term, since $D_{i,L}(t)a$ is $s_D$-sparse, it takes $O(ms_D)$-time. For the second term, we have

$$\Delta W_L(t)^\top (D_{i,L}(t)a) = \sum_{j=1}^{r_t} z_{L,j} y_{L,j}^\top (D_{i,L}(t)a) = \sum_{j=1}^{r_t} z_{L,j} \cdot \langle y_{L,j}, D_{i,L}(t)a \rangle.$$

Each inner-product takes $O(s_D)$-time and it takes $O(mr_t + s_D r_t) = O(mr_t)$-time in total. Hence, in $O(m(s_D + r_t))$-time, we compute the vector $W_L(t)^\top D_{i,L}(t)a$. Note that we do not assume the sparsity of $a$.

Thus, by repeating this process for the $L - \ell$ intermediate matrices $W_j^\top(t) D_{i,j}(t)$, we can obtain the vector

$$\left( \prod_{j=\ell+1}^{L} W_j^\top(t) D_{i,j}(t) \right) a$$

in time $O((L - \ell)m(s_D + r_t))$. Finally, by multiplying a sparse diagonal matrix $D_{i,\ell}(t)$, we get the desired vector $u_{i,\ell}(t)$.

**Compute $v_{i,\ell}(t)$.** Note that $v_{i,\ell}(t)$ is essentially $h_{i,\ell-1}(t)$, so we consider how to compute $h_{i,\ell}(t)$ for general $\ell \in [L]$. Recall that

$$h_{i,\ell}(t) = \phi((W_\ell(0) + \Delta W_\ell(t))h_{i,\ell-1}(t)),$$

since $h_{i,\ell-1}(t)$ is $s_D$-sparse, the product $W_\ell(0h_{i,\ell-1}(t)))$ can be computed in $O(ms_D)$ time. For the product $\Delta W_\ell(t)h_{i,\ell-1}(t)$ can be computed use the low rank decomposition, which takes $O(mr_t)$ time. Apply the threshold ReLU takes $O(m)$ time. Hence, the total time is $O(m(r_t + s_D))$ time.

The running time results are summarized in the following lemma:

**Lemma C.4.** *For $\ell \in [L]$ and $i \in [n]$, suppose $\|D_{i,\ell}(0)\|_0 \leq k$. Let $t > 0$. Suppose the change of $D_{i,\ell}$ is sparse, i.e., $\|D_{i,\ell}(t) - D_{i,\ell}(0)\|_0 \leq s$. For $\ell \in [L]$, $i \in [n]$, for any $t > 0$, suppose the change of $W_\ell$ is of low-rank, i.e., $\Delta W_\ell(t) = \sum_{j=1}^{r_t} y_{\ell,j} z_{\ell,j}^\top$. We further assume that $\{y_{\ell,j}, z_{\ell,j}\}_{\ell \in [L], j \in [r_t]}$ and $\{W_1(0)x_i\}_{i \in [n]}$ are pre-computed.*

*Then, for any $\ell \in [L]$ and $i \in [n]$, the vectors $u_{\ell,i}(t), v_{\ell,i}(t) \in \mathbb{R}^m$ can be computed in $O(mL(s + k + r_t))$-time.*

As a direct consequence, if we combine Lemma C.2 and Lemma C.4, then we get the following corollary:

**Corollary C.5.** *For $\ell \in [L]$ and $i \in [n]$, we can compute $v_{i,\ell}(t), u_{i,\ell}(t) \in \mathbb{R}^m$ as in Algorithm 2 with the following time bound:*

- *Compute $u_{i,\ell}(t)$ in time $O(mL(s + k + r_\ell))$.*

- *Compute $v_{i,\ell}(t)$ in time $O(m(s + k + r_\ell))$.*

# D    FAST TENSOR PRODUCT REGRESSION

In this section, we design a generic algorithm for solving the following type of regression task:

Given two matrices $U = [u_1^\top, \ldots, u_n^\top]^\top, V = [v_1^\top, \ldots, v_n^\top] \in \mathbb{R}^{m \times n}$ with $m \gg n$, consider the matrix $J \in \mathbb{R}^{n \times m^2}$ formed by

$$J = \begin{bmatrix} \text{vec}(u_1 v_1^\top)^\top \\ \text{vec}(u_2 v_2^\top)^\top \\ \vdots \\ \text{vec}(u_n v_n^\top)^\top \end{bmatrix}.$$

We are also given a vector $c \in \mathbb{R}^n$, the goal is to solve the following regression task:

$$\min_{x \in \mathbb{R}^n} \|JJ^\top x - c\|_2^2.$$

Our main theorem for this section is as follows:

**Theorem D.1** (Restatement of Theorem D.14). *Given two $n \times m$ matrices $U$ and $V$, and a target vector $c \in \mathbb{R}^n$. Let $J = [\text{vec}(u_1 v_1^\top)^\top, \ldots, \text{vec}(u_n v_n^\top)^\top] \in \mathbb{R}^{n \times m^2}$. There is an algorithm (Algorithm 6) takes $\widetilde{O}(nm + n^2(\log(\kappa/\epsilon) + \log(m/\epsilon\delta)\epsilon^{-2}) + n^\omega)$ time and outputs a vector $\widehat{x} \in \mathbb{R}^n$ such that*

$$\|JJ^\top \widehat{x} - c\|_2 \leq \epsilon \|c\|_2$$

*holds with probability at least $1 - \delta$, and $\kappa$ is the condition number of $J$.*

## D.1    APPROXIMATE $J$ VIA TensorSketch

We introduce the notion of TensorSketch for two vectors:

**Definition D.2.** *Let $h_1, h_2 : [m] \to [s]$ be 3-wise independent hash functions, also let $\sigma : [m] \to \{\pm 1\}$ be a 4-wise independent random sign function. The degree two* TensorSketch *transform, $S : \mathbb{R}^m \times \mathbb{R}^m \to \mathbb{R}^s$ is defined as follows: for any $i, j \in [m]$ and $r \in [s]$,*

$$S_{r,(i,j)} = \sigma_1(i) \cdot \sigma_2(j) \cdot \mathbf{1}[h_1(i) + h_2(j) = r \bmod s].$$

**Remark D.3.** *Apply $S$ to two vectors $x, y \in \mathbb{R}^m$ can be implemented in time $O(s \log s + \mathrm{nnz}(x) + \mathrm{nnz}(y))$.*

We introduce one key technical lemma from Avron et al. (2014):

**Lemma D.4** (Theorem 1 of Avron et al. (2014)). *Let $S \in \mathbb{R}^{s \times m^2}$ be the TensorSketch matrix, consider a fixed $n$-dimensional subspace $V$. If $s = \Omega(n^2/(\epsilon^2 \delta))$, then with probability at least $1 - \delta$, $\|Sx\|_2 = (1 \pm \epsilon)\|x\|_2$ simultaneously for all $x \in V$.*

Now we are ready to prove the main lemma of this section:

**Lemma D.5.** *Let $\epsilon, \delta \in (0, 1)$ denote two parameters. Let $J \in \mathbb{R}^{n \times m^2}$ be a matrix such that $i$-th row of $J$ is defined as $\mathrm{vec}(u_i v_i^\top)$ for some $u_i, v_i \in \mathbb{R}^m$. Then, we can compute a matrix $\widetilde{J} \in \mathbb{R}^{n \times s}$ such that for any vector $x \in \mathbb{R}^n$, with probability at least $1 - \delta$, we have*

$$\|\widetilde{J}^\top x\|_2 = (1 \pm \epsilon)\|J^\top x\|_2,$$

*where $s = \Omega(n^2/(\epsilon^2 \delta))$. The time to compute $\widetilde{J}$ is $O(ns \log s + \mathrm{nnz}(U) + \mathrm{nnz}(V))$.*

*Proof.* Notice that the row space of matrix $J$ can be viewed as an $n$-dimensional subspace, hence, by Lemma D.4, the TensorSketch matrix $S$ with $s = \Omega(n^2/(\epsilon^2 \delta))$ can preserve the length of all vectors in the subspace generated by $J^\top$ with probability $1 - \delta$, to a multiplicative factor of $1 \pm \epsilon$.

The running time part is to apply the FFT algorithm to each row of $J$ with a total of $n$ rows. For each row, it takes $O(s \log s + m)$ time, hence the overall running time is $O(n(s \log s + m))$. $\square$

## D.2   Approximate $J$ via TensorSRHT

We note that the dependence on the target dimension of sketching is $O(1/\delta)$ for TensorSketch. We introduce another kind of sketching technique for tensor, called TensorSRHT. The tradeoff is we lose input sparsity runtime of matrices $U$ and $V$.

**Definition D.6.** *We define the TensorSRHT $S : \mathbb{R}^d \times \mathbb{R}^d \to \mathbb{R}^m$ as $S = \frac{1}{\sqrt{m}} P \cdot (HD_1 \times HD_2)$, where each row of $P \in \{0, 1\}^{m \times d^2}$ contains only one $1$ at a random coordinate, one can view $P$ as a sampling matrix. $H$ is a $d \times d$ Hadamard matrix, and $D_1, D_2$ are two $d \times d$ independent diagonal matrices with diagonals that are each independently set to be a Rademacher random variable (uniform in $\{-1, 1\}$).*

**Remark D.7.** *By using FFT algorithm, apply $S$ to two vectors $x, y \in \mathbb{R}^m$ takes time $O(m \log m + s)$.*

We again introduce a technical lemma for TensorSRHT.

**Lemma D.8** (Theorem 3 of Ahle et al. (2020)). *Let $S \in \mathbb{R}^{s \times m^2}$ be the TensorSRHT matrix, consider a fixed $n$-dimensional subspace $V$. If $s = \Omega(n \log^3(nm/\epsilon\delta)\epsilon^{-2})$, then with probability at least $1 - \delta$, $\|Sx\|_2 = (1 \pm \epsilon)\|x\|_2$ simultaneously for all $x \in V$.*

**Lemma D.9.** *Let $\epsilon, \delta \in (0, 1)$ denote two parameters. Given a list of vectors $u_1, \cdots, u_m, v_1, \cdots, v_m \in \mathbb{R}^m$. Let $J \in \mathbb{R}^{n \times m^2}$ be a matrix with $i$-th row of $J$ is defined as $\mathrm{vec}(u_i v_i^\top)$. Then, we can compute a matrix $\widetilde{J} \in \mathbb{R}^{n \times s}$ such that for any vector $x \in \mathbb{R}^n$, with probability at least $1 - \delta$, we have*

$$\|\widetilde{J}^\top x\|_2 = (1 \pm \epsilon)\|J^\top x\|_2,$$

*where $s = \Omega(n \log^3(nm/(\epsilon\delta))\epsilon^{-2})$. The time to compute $\widetilde{J}$ is $O(n(m \log m + s))$.*

*Proof.* The correctness follows directly from Lemma D.8. The running time follows from the FFT algorithm to each row of $J$, each application takes $O(m \log m + s)$ time, and we need to apply it to $n$ rows. $\square$

### D.3 TENSOR TRICK AS A PRECONDITIONER

In this section, we use TensorSketch and TensorSRHT as a preconditioner to solve a regression task involving $JJ^\top$. This is an important step to implement the Gram-Newton-Gauss iteration Cai et al. (2019); Brand et al. (2021).

Before proceeding, we introduce the notion of *subspace embedding*:

**Definition D.10** (Subspace Embedding, Sarlos (2006)). *Let $A \in \mathbb{R}^{N \times k}$, we say a matrix $S \in \mathbb{R}^{s \times N}$ is a $(1 \pm \epsilon) - \ell_2$ subspace embedding for $A$ if for any $x \in \mathbb{R}^k$, we have $\|SAx\|_2^2 = (1 \pm \epsilon)\|Ax\|_2^2$. Equivalently, $\|I - U^\top S^\top SU\| \le \epsilon$ where $U$ is an orthonormal basis for the column space of $A$.*

We will mainly utilize efficient subspace embedding.

**Definition D.11** (Lu et al. (2013); Woodruff (2014)). *Given a matrix $A \in \mathbb{R}^{N \times k}$ with $N = \mathrm{poly}(k)$, then we can compute an $S \in \mathbb{R}^{k\mathrm{poly}(\log(k/\delta))/\epsilon^2 \times N}$ such that with probability at least $1 - \delta$, we have*

$$\|SAx\|_2 = (1 \pm \epsilon)\|Ax\|_2$$

*hols for all $x \in \mathbb{R}^k$. Moreover, $SA$ can be computed in $O(Nk \log((k \log N)/\epsilon))$ time.*

---

**Algorithm 5** Fast Regression algorithm of Brand et al. (2021)]

---

1: **procedure** FASTREGRESSION($A, y, \epsilon$)                         ▷ Lemma D.12
2:                                     ▷ $A \in \mathbb{R}^{N \times k}$ is full rank, $\epsilon \in (0, 1/2)$
3:     Compute a subspace embedding $SA$                 ▷ $S \in \mathbb{R}^{k\mathrm{poly}(\log k) \times N}$
4:     Compute $R$ such that $SAR$ has orthonormal columns via QR decomposition   ▷ $R \in \mathbb{R}^{k \times k}$
5:     $z_0 = \mathbf{0}_k \in \mathbb{R}^k$
6:     $t \leftarrow 0$
7:     **while** $\|A^\top A R z_t - y\|_2 \ge \epsilon$ **do**
8:         $z_{t+1} \leftarrow z_t - (R^\top A^\top A R)^\top (R^\top A^\top A R z_t - R^\top y)$
9:         $t \leftarrow t + 1$
10:     **end while**
11:     **return** $R z_t$
12: **end procedure**

---

**Lemma D.12** (Lemma 4.2 of Brand et al. (2021)). *Let $N = \Omega(k\mathrm{poly}(\log k))$. Given a matrix $A \in \mathbb{R}^{N \times k}$, let $\kappa$ denote its condition number. Consider the following regression task:*

$$\min_{x \in \mathbb{R}^k} \|A^\top A x - y\|_2.$$

*Using the procedure* FASTREGRESSION *(Algorithm 5), with probability at least $1 - \delta$, we can compute an $\epsilon$-approximate solution $\widehat{x}$ satisfying*

$$\|A^\top A \widehat{x} - y\|_2 \le \epsilon \|y\|_2$$

*in time $\widetilde{O}(Nk \log(\kappa/\epsilon) + k^\omega)$.*

Our algorithm is similar to the ridge regression procedure in Song et al. (2021), where they first apply their sketching algorithm as a bootstrapping to reduce the dimension of the original matrix, then use another subspace embedding to proceed and get stronger guarantee.

We shall first prove a useful lemma.

**Lemma D.13.** *Let $A \in \mathbb{R}^{N \times k}$, suppose $SA$ is a subspace embedding for $A$ (Def. D.11), then we have for any $x \in \mathbb{R}^k$, with probability at least $1 - \delta$,*

$$\|(SA)^\top SAx - b\|_2 = (1 \pm \epsilon)\|A^\top A x - b\|_2.$$

*Proof.* Throughout the proof, we condition on the event that $S$ preserves the length of all vectors in the column space of $A$.

Note that

$$\|(SA)^\top SAx - b\|_2^2 = \|(SA)^\top SAx\|_2^2 + \|b\|_2^2 - 2\langle (SA)^\top SAx, b \rangle.$$

We will first bound the norm of $(SA)^\top SAx$, then the inner product term.

**Bounding** $\|(SA)^\top SAx\|_2^2$

Let $U \in \mathbb{R}^{N \times k}$ be an orthonormal basis of $A$, then use the equivalent definition of subspace embedding, we have $\|U^\top S^\top SU - I\| \leq \epsilon$, this means all the eigenvalues of $U^\top S^\top SU$ lie in the range of of $[(1-\epsilon)^2, (1+\epsilon)^2]$. Let $V$ denote the matrix $U^\top S^\top SU$, then we know that all eigenvalues of $V^\top V$ lie in range $[(1-\epsilon)^4, (1+\epsilon)^4]$. Setting $\epsilon$ as $\epsilon/4$, we arrive at $\|V^\top V - I\| \leq \epsilon$. This shows that for any $x \in \mathbb{R}^k$, we have $\|(SA)^\top SAx\|_2 = (1 \pm \epsilon)\|A^\top Ax\|_2$.

**Bounding** $\langle (SA)^\top SAx, b \rangle$ Note that

$$
\begin{aligned}
\langle (SA)^\top SAx, b \rangle &= \langle SAx, SAb \rangle \\
&= 1/2 \cdot (\|SAx\|_2^2 + \|SAb\|_2^2 - \|SA(x-b)\|_2^2) \\
&= 1/2 \cdot (1 \pm \epsilon)(\|Ax\|_2^2 + \|Ab\|_2^2 - \|A(x-b)\|_2^2) \\
&= (1 \pm \epsilon)\langle A^\top Ax, b \rangle.
\end{aligned}
$$

Combining these two terms, we conclude that, with probability at least $1 - \delta$,

$$
\|(SA)^\top SAx - b\|_2 = (1 \pm \epsilon)\|A^\top Ax - b\|_2.
$$

$\square$

---

**Algorithm 6** Fast Regression via tensor trick

---

1: **procedure** FASTTENSORREGRESSION($\{u_i\}_{i=1}^n \in \mathbb{R}^{m \times n}, \{v_i\}_{i=1}^n \in \mathbb{R}^{m \times n}, c \in \mathbb{R}^n$)  $\triangleright$ Theorem D.14
2: $\triangleright J = [\text{vec}(u_1 v_1^\top)^\top, \ \text{vec}(u_2 v_2^\top)^\top, \ldots, \ \text{vec}(u_n v_n^\top)^\top]^\top \in \mathbb{R}^{n \times m^2}$
3: $\quad s_1 \leftarrow \Theta(n \log^3(nm/(\epsilon\delta))\epsilon^{-2})$
4: $\quad s_2 \leftarrow \Theta((n + \log m) \log n)$
5: $\quad$ Let $S_1 \in \mathbb{R}^{s_1 \times m^2}$ be a sketching matrix  $\triangleright S_1$ can be TensorSketch or TensorSRHT
6: $\quad$ Compute $\widetilde{J} = J S_1^\top$ via FFT algorithm  $\triangleright \widetilde{J} \in \mathbb{R}^{n \times s_1}$
7: $\quad$ Let $S_2 \in \mathbb{R}^{s_2 \times s_1}$ be a sketching matrix defined in Definition D.11
8: $\quad$ Compute a subspace embedding $S_2 \widetilde{J}^\top$
9: $\quad$ Compute $R$ such that $S_2 \widetilde{J}^\top R$ has orthonormal columns via QR decomposition  $\triangleright R \in \mathbb{R}^{n \times n}$
10: $\quad z_0 \leftarrow \mathbf{0}_k \in \mathbb{R}^k$
11: $\quad t \leftarrow 0$
12: $\quad$ **while** $\|\widetilde{J}\widetilde{J}^\top R z_t - c\|_2 \geq \epsilon$ **do**
13: $\quad\quad z_{t+1} \leftarrow z_t - (R^\top \widetilde{J}\widetilde{J}^\top R)^\top (R^\top \widetilde{J}\widetilde{J}^\top R z_t - R^\top c)$
14: $\quad\quad t \leftarrow t + 1$
15: $\quad$ **end while**
16: $\quad$ **return** $R z_t$
17: **end procedure**

---

**Theorem D.14.** *Given two $n \times m$ matrices $U$ and $V$, and a target vector $c \in \mathbb{R}^n$. Let $J = [\text{vec}(u_1 v_1^\top)^\top, \ldots, \text{vec}(u_n v_n^\top)^\top] \in \mathbb{R}^{n \times m^2}$. There is an algorithm (Algorithm 6) takes $\widetilde{O}(nm + n^2(\log(\kappa/\epsilon) + \log(m/\delta)) + n^\omega)$ time and outputs a vector $\widehat{x} \in \mathbb{R}^n$ such that*

$$
\|JJ^\top \widehat{x} - c\|_2 \leq \epsilon \|c\|_2
$$

*holds with probability at least $1 - \delta$, and $\kappa$ is the condition number of $J$.*

*Proof.* We can decompose Algorithm 6 into two parts:

- Applying $S_1$ to efficiently form matrix $\widetilde{J}$ to approximate $J$ and reduce its dimension, notice here we only need $\epsilon$ for this part to be a small constant, pick $\epsilon = 0.1$ suffices.

- Using $S_2$ as a preconditioner and solve the regression problem iteratively.

Let $\widehat{x}$ denote the solution found by the iterative regime. We will prove this statement in two-folds:

- First, we will show that $\|\widetilde{J}\widetilde{J}^\top \widehat{x} - c\|_2 \leq \epsilon\|c\|_2$ with probability at least $1 - \delta$;

- Then, we will show that $\|JJ^\top \widehat{x} - c\|_2 = (1 \pm 0.1)\|\widetilde{J}\widetilde{J}^\top \widehat{x} - c\|_2$ with probability at least $1 - \delta$.

Combining these two statements, we can show that

$$\|JJ^\top \widehat{x} - c\|_2 = (1 \pm 0.1)\|\widetilde{J}\widetilde{J}^\top \widehat{x} - c\|_2$$
$$\leq 1.1\epsilon\|c\|_2$$

Setting $\epsilon$ to $\epsilon/1.1$ and $\delta$ to $\delta/2$, we conclude our proof. It remains to prove these two parts.

**Part 1.** $\|\widetilde{J}\widetilde{J}^\top \widehat{x} - c\|_2 \leq \epsilon\|c\|_2$ We observe the iterative procedure is essentially the same as running FASTREGRESSION on input $\widetilde{J}^\top, y, \epsilon$, hence by Lemma D.12, we have with probability at least $1 - \delta$, $\|\widetilde{J}\widetilde{J}^\top \widehat{x} - c\|_2 \leq \epsilon\|c\|_2$.

**Part 2.** $\|JJ^\top \widehat{x} - c\|_2 = (1 \pm 0.1)\|\widetilde{J}\widetilde{J}^\top \widehat{x} - c\|_2$ To prove this part, note that by Lemma D.8, we know that $\widetilde{J}^\top$ is a subspace embedding for $J^\top$. Hence, we can utilize Lemma D.13 and get that, with probability at least $1 - \delta$, we have $\|JJ^\top \widehat{x} - c\|_2 = (1 \pm 0.1)\|\widetilde{J}\widetilde{J}^\top \widehat{x} - c\|_2$.

Combining these two parts, we have proven the correctness of the theorem. It remains to justify the running time. Note that running time can be decomposed into two parts: 1). The time to generate $\widetilde{J}$, 2). The time to compute $\widehat{x}$ via iterative scheme.

**Part 1. Generate $\widetilde{J}$** To generate $\widetilde{J}$, we apply $S_1 \in \mathbb{R}^{s_1 \times m^2}$ which is a TensorSRHT. By Lemma D.9, it takes $O(n(m \log m + s_1))$ time to compute $\widetilde{J}$, plug in $s_1 = \Theta(n \log^3(nm/\delta))$, the time is $\widetilde{O}(nm)$.

**Part 2. Compute $\widehat{x}$** To compute $\widehat{x}$, essentially we run FASTREGRESSION on $\widetilde{J}^\top, c, \epsilon$, hence by Lemma D.12, it takes $\widetilde{O}(s_2 n \log(\kappa/\epsilon) + n^\omega)$ time, with $s_2 = \Theta((n + \log m) \log n)$ and $\kappa$ is the condition number of $\widetilde{J}$, which has the guarantee $\kappa = (1 \pm \epsilon)\kappa(J)$. Hence, the overall running time of this part is $\widetilde{O}(n^2 \log(\kappa/\epsilon) + n^\omega)$.

Put things together, the overall running time is $\widetilde{O}(nm + n^2 \log(\kappa/\epsilon) + n^\omega)$. $\qquad\square$

**Remark D.15.** *Due to the probability requirement (union bounding over all layers and all data points), here we only prove by using* TensorSRHT. *One can use similar strategy to obtain an input sparsity time version using* TensorSketch. *We remark that this framework is similar to the approach* Song et al. (2021) *takes to solve kernel ridge regression, where one first uses a shallow but fast sketch to bootstrap, then use another sketching to proceed with the main task.*

# E SPECTRAL PROPERTIES OF OVER-PARAMETRIZED DEEP NEURAL NETWORK

In this section, we study the spectral properties of our Gram matrix and connects it to multi-layer NTK.

## E.1 BOUNDS ON THE LEAST EIGENVALUE OF KERNEL AT INITIALIZATION

We first define the Gram matrices for multiple layer neural network.

**Definition E.1** (Multiple layer Gram matrix)**.** *The Gram matrices* $\mathbf{K}_\ell \in \mathbb{R}^{n \times n}$ *for* $\ell \in \{0, \dots, L\}$ *of an L-layer neural network are defined as follows:*

- $(\mathbf{K}_0)_{i,j} := x_i^\top x_j$

- *For $\ell > 0$, let $\Sigma_{\ell,i,j} := \begin{bmatrix} (\mathbf{K}_{\ell-1})_{i,i} & (\mathbf{K}_{\ell-1})_{i,j} \\ (\mathbf{K}_{\ell-1})_{j,i} & (\mathbf{K}_{\ell-1})_{j,j} \end{bmatrix} \in \mathbb{R}^{2\times 2}$ for any $(i,j) \in [n] \times [n]$. Then,*

$$(\mathbf{K}_\ell)_{i,j} := \mathop{\mathbb{E}}_{(x_1,x_2)\sim\mathcal{N}(\mathbf{0},2\Sigma_{\ell-1,i,j})}[\phi(x_1)\phi(x_2)] \quad \forall \ell \in [L-1],$$

$$(\mathbf{K}_L)_{i,j} := \mathop{\mathbb{E}}_{(x_1,x_2)\sim\mathcal{N}(\mathbf{0},2\Sigma_{L-1,i,j})}[\phi'(x_1)\phi'(x_2)]$$

*Let $\lambda_L := \lambda_{\min}(\mathbf{K}_L)$ to be the minimum eigenvalue of the NTK kernel $\mathbf{K}_L$.*

In the following lemma, we generalize Lemma C.3 in Brand et al. (2021) (also Lemma 3 in Cai et al. (2019)) into multiple layer neural networks.

**Lemma E.2** (Bounds on the least eigenvalue at initialization, multiple layer version of Lemma C.3 in Brand et al. (2021)). *Let $\lambda_\ell$ denote the minimum eigenvalue of NTK defined for $\ell$-th layer of neural networks. Suppose $m_\ell = \Omega(\lambda_\ell^{-2} n^2 \log(n/\delta))$, then with probability $1 - \delta$, we have*

$$\lambda_{\min}(G_\ell(0)) \geq \frac{3}{4}\lambda_\ell, \quad \forall \ell \in [L]$$

*Proof.* For any $\ell \in [L]$, we have $G_\ell = J_\ell J_\ell^\top \in \mathbb{R}^{n\times n}$. Hence, for any $i,j \in [n]$,

$$(G_\ell)_{i,j} = \mathrm{vec}(\frac{\partial f(W,x_i)}{\partial W_\ell})^\top \mathrm{vec}(\frac{\partial f(W,x_j)}{\partial W_\ell})$$

$$= \mathrm{vec}\left(D_{i,\ell}\prod_{k=\ell+1}^{L} W_k^\top D_{i,k} a h_{i,\ell-1}^\top\right)^\top \mathrm{vec}\left(D_{j,\ell}\prod_{k=\ell+1}^{L} W_k^\top D_{j,k} a h_{j,\ell-1}^\top\right)$$

$$= \left((h_{i,\ell-1}\otimes I_{m_\ell})\left(D_{i,\ell}\prod_{k=\ell+1}^{L} W_k^\top D_{i,k} a\right)\right)^\top (h_{j,\ell-1}\otimes I_{m_\ell})\left(D_{j,\ell}\prod_{k=\ell+1}^{L} W_k^\top D_{j,k} a\right)$$

$$= \left(D_{i,\ell}\prod_{k=\ell+1}^{L} W_k^\top D_{i,k} a\right)^\top (h_{i,\ell-1}^\top\otimes I_{m_\ell})(h_{j,\ell-1}\otimes I_{m_\ell})\left(D_{j,\ell}\prod_{k=\ell+1}^{L} W_k^\top D_{j,k} a\right)$$

$$= a^\top\left(D_{i,\ell}\prod_{k=\ell+1}^{L} W_k^\top D_{i,k}\right)^\top\left(D_{j,\ell}\prod_{k=\ell+1}^{L} W_k^\top D_{j,k}\right) a \cdot h_{i,\ell-1}^\top h_{j,\ell-1},$$

where

$$h_{i,\ell-1} = \prod_{k=1}^{\ell-1} D_{i,k} W_k x_i.$$

In particular,

$$(G_L)_{i,j} = a^\top D_{i,L} D_{j,L} a \cdot h_{i,L-1}^\top h_{j,L-1}$$

$$= \frac{1}{m}\sum_{r=1}^{m} \phi'(\langle w_L^{(r)}, h_{i,L-1}\rangle)\phi'(\langle w_L^{(r)}, h_{j,L-1}\rangle)h_{i,L-1}^\top h_{j,L-1} \tag{4}$$

We will prove that $\|G_L - \mathbf{K}_L\|_\infty$ is small, which implies that $\lambda_{\min}(G_L)$ is close to $\lambda_\ell$. The proof idea is similar to Du et al. (2019a) via induction on $\ell$.

For $\ell = 1$, recall $(g_{1,i})_k = \sum_{b\in[m]}(W_1)_{k,b}(x_i)_b$ for $k \in [m]$. Hence, for any $k \in [m]$,

$$\mathbb{E}[(g_{1,i})_k (g_{1,j})_k] = \sum_{b,b'\in[m]} \mathbb{E}[(W_1)_{k,b}(W_1)_{k,b'}(x_i)_b(x_j)_{b'}]$$

$$= \sum_{b\in[m]} \mathbb{E}[(W_1)_{k,b}^2]\cdot(x_i)_b(x_j)_b \qquad ((W_1)_{k,b}\sim\mathcal{N}(0,\tfrac{2}{m}).)$$

$$= \frac{2}{m}\sum_{b\in[m]}(x_i)_b(x_j)_b$$

$$= \frac{2}{m}x_i^\top x_j.$$

Then, we have

$$
\begin{aligned}
\mathbb{E}[h_{1,i}^\top h_{1,j}] &= \sum_{k \in [m]} \mathbb{E}[(h_{1,i})_k (h_{1,j})_k] \\
&= \sum_{k \in [m]} \mathbb{E}[\phi((g_{1,i})_k)\phi((g_{1,j})_k)] \\
&= \sum_{k \in [m]} \mathbb{E}_{(u,v) \sim \mathcal{N}(0, \frac{2}{m}\Sigma_{1,i,j})}[\phi(u)\phi(v)] \\
&= \mathbb{E}_{(u,v) \sim \mathcal{N}(0, \frac{2}{m}\Sigma_{1,i,j})}[m\phi(u)\phi(v)] \\
&= \mathbb{E}_{(u',v') \sim \mathcal{N}(0, 2\Sigma_{1,i,j})}[\phi(u')\phi(v')] \\
&= (\mathbf{K}_1)_{i,j}.
\end{aligned}
$$

Next, we will show that $h_{1,i}^\top h_{1,j}$ concentrates around its expectation. First, for any $k \in [m]$,

$$
|(h_{1,i})_k (h_{1,j})_k| \leq |(g_{1,i})_k (g_{1,j})_k| \leq |\langle (W_1)_{k,*}, x_i \rangle| \cdot |\langle (W_1)_{k,*}, x_j \rangle|.
$$

Since $\langle (W_1)_{k,*}, x_i \rangle \sim \mathcal{N}(0, \frac{2\|x_i\|_2^2}{m})$, by the concentration of Gaussian distribution,

$$
|\langle (W_1)_{k,*}, x_i \rangle| \leq \sqrt{c} \quad \forall k \in [m], i \in [n]
$$

holds with probability at least $1 - mne^{-cm/4}$. Conditioned on this event, we have $|(h_{1,i})_k (h_{1,j})_k| \leq c$ for all $i, j \in [n]$ and $k \in [m]$. Then, by Hoeffding's inequality, we have for any $(i,j) \in [n] \times [n]$,

$$
\Pr\left[ |h_{1,i}^\top h_{1,j} - (\mathbf{K}_1)_{i,j}| \geq t \right] \leq \exp\left( -\frac{t^2}{2m \cdot (2c)^2} \right) = \exp(-\Omega(t^2/(mc^2))).
$$

Hence, by union bound, we get that

$$
\max_{(i,j) \in [n] \times [n]} |h_{1,i}^\top h_{1,j} - (\mathbf{K}_1)_{i,j}| \leq t
$$

with probability at least

$$
1 - mn\exp(-\Omega(mc)) - n^2 \exp(-\Omega(t^2/(mc^2))).
$$

If we choose $c := \frac{\log(mnL/\delta)}{m}$ and $t := m^{-1/2} \cdot \mathrm{polylog}(nL/\delta)$, we have with probability at least $1 - \frac{\delta}{L}$,

$$
\max_{(i,j) \in [n] \times [n]} |h_{1,i}^\top h_{1,j} - (\mathbf{K}_1)_{i,j}| \leq \widetilde{O}(m^{-1/2}).
$$

Suppose for $\ell = 1, \ldots h$ ($h < L$), we have

$$
\max_{(i,j) \in [n] \times [n]} |h_{\ell,i}^\top h_{\ell,j} - (\mathbf{K}_\ell)_{i,j}| \leq \widetilde{O}(m^{-1/2}).
$$

Consider $\ell = h + 1$. By a similar computation, we have

$$
\mathbb{E}_{W_\ell}[(g_{\ell,i})_k (g_{\ell,j})_k] = \frac{2}{m} h_{\ell-1,i}^\top h_{\ell-1,j}.
$$

Define a new covariance matrix

$$
\widehat{\Sigma}_{\ell,i,j} := \begin{bmatrix} h_{\ell-1,i}^\top h_{\ell-1,i} & h_{\ell-1,i}^\top h_{\ell-1,j} \\ h_{\ell-1,j}^\top h_{\ell-1,i} & h_{\ell-1,j}^\top h_{\ell-1,j} \end{bmatrix} \quad \forall (i,j) \in [n] \times [n].
$$

We have

$$
\begin{aligned}
\mathbb{E}_{W_\ell}[h_{i,\ell}^\top h_{j,\ell}] &= \sum_{k \in [m]} \mathbb{E}_{(u,v) \sim \mathcal{N}(0, \frac{2}{m}\widehat{\Sigma}_{\ell,i,j})}[\phi(u)\phi(v)] \\
&= \mathbb{E}_{(u',v') \sim \mathcal{N}(0, 2\widehat{\Sigma}_{\ell,i,j})}[\phi(u')\phi(v')] \\
&:= (\widehat{\mathbf{K}}_\ell)_{i,j}.
\end{aligned}
$$

Hence, we have with probability at least $1 - \frac{\delta}{L}$,

$$\max_{(i,j)\in[n]\times[n]} \left| h_{\ell,i}^\top h_{\ell,j} - (\widehat{\mathbf{K}}_\ell)_{i,j} \right| \le \widetilde{O}(m^{-1/2}). \tag{5}$$

It remains to upper bound the difference $\|\widehat{\mathbf{K}}_\ell - \mathbf{K}_\ell\|_\infty$.

$$\left\|\widehat{\mathbf{K}}_\ell - \mathbf{K}_\ell\right\|_\infty = \max_{(i,j)\in[n]\times[n]} \left| \mathop{\mathbb{E}}_{(u,v)\sim\mathcal{N}(0,2\widehat{\Sigma}_{\ell,i,j})} [\phi(u)\phi(v)] - \mathop{\mathbb{E}}_{(u,v)\sim\mathcal{N}(0,2\Sigma_{\ell,i,j})} [\phi(u)\phi(v)] \right|.$$

Recall that

$$\Sigma_{\ell,i,j} := \begin{bmatrix} (\mathbf{K}_{\ell-1})_{i,i} & (\mathbf{K}_{\ell-1})_{i,j} \\ (\mathbf{K}_{\ell-1})_{j,i} & (\mathbf{K}_{\ell-1})_{j,j} \end{bmatrix} \quad \forall(i,j)\in[n]\times[n],$$

and hence, by the induction hypothesis, we have

$$\|\widehat{\Sigma}_{\ell,i,j} - \Sigma_{\ell,i,j}\|_\infty \le \max_{(i,j)\in[n]\times[n]} \left| h_{\ell-1,i}^\top h_{\ell-1,j} - (\mathbf{K}_{\ell-1})_{i,j} \right| = \widetilde{O}(m^{-1/2}).$$

Notice that $\widehat{\Sigma}_{\ell,i,j}$ can be written as

$$\begin{bmatrix} \|h_{\ell-1,i}\|_2^2 & \cos(\theta_{\ell,i,j})\|h_{\ell-1,i}\|_2\|h_{\ell-1,j}\|_2 \\ \cos(\theta_{\ell,i,j})\|h_{\ell-1,i}\|_2\|h_{\ell-1,j}\|_2 & \|h_{\ell-1,j}\|_2^2 \end{bmatrix}.$$

Moreover, when $\phi$ is the ReLU function, we have

$$\mathop{\mathbb{E}}_{(u,v)\sim\mathcal{N}(0,2\widehat{\Sigma}_{\ell,i,j})} [\phi(u)\phi(v)] = 2\|h_{\ell-1,i}\|_2\|h_{\ell-1,j}\|_2 \cdot F(\theta_{\ell,i,j}),$$

where

$$F(\theta) := \mathop{\mathbb{E}}_{(u,v)\sim\mathcal{N}(0,\Sigma(\theta))} [\phi(u)\phi(v)] \quad \text{with} \quad \Sigma(\theta) := \begin{bmatrix} 1 & \cos(\theta) \\ \cos(\theta) & 1 \end{bmatrix}.$$

We note that $F(\theta)$ has the following analytic form:

$$F(\theta) = \frac{1}{2\pi}(\sin(\theta) + (\pi - \theta)\cos(\theta)) \in [0, 1/2]. \tag{6}$$

Similarly,

$$\mathop{\mathbb{E}}_{(u,v)\sim\mathcal{N}(0,2\Sigma_{\ell,i,j})} [\phi(u)\phi(v)] = 2\sqrt{(\mathbf{K}_{\ell-1})_{i,i}(\mathbf{K}_{\ell-1})_{j,j}} \cdot F(\tau_{\ell,i,j}),$$

where $\tau_{\ell,i,j} := \cos^{-1}\left(\frac{(\mathbf{K}_{\ell-1})_{i,j}}{\sqrt{(\mathbf{K}_{\ell-1})_{i,i}(\mathbf{K}_{\ell-1})_{j,j}}}\right)$. By the induction hypothesis, we have $(\mathbf{K}_\ell)_{i,j} \in h_{\ell,i}^\top h_{\ell,j} \pm \widetilde{O}(m^{-1/2})$ for all $i,j \in [n]$. By Lemma F.7, we also have $\|h_{\ell,i}\|_2 \in 1\pm\epsilon$ for all $\ell \in [L]$ and $i \in [n]$ with probability $1 - O(nL)\cdot e^{-\Omega(m\epsilon^2/L)}$. They implies that $\cos(\tau_{\ell,i,j}) \in \cos(\theta) \pm \widetilde{O}(m^{-1/2})$. Thus, by Taylor's theorem, we have

$$|F(\theta_{\ell,i,j}) - F(\tau_{\ell,i,j})| \le \widetilde{O}(m^{-1/2}).$$

Therefore, we have

$$\left| \mathop{\mathbb{E}}_{(u,v)\sim\mathcal{N}(0,2\widehat{\Sigma}_{\ell,i,j})} [\phi(u)\phi(v)] - \mathop{\mathbb{E}}_{(u,v)\sim\mathcal{N}(0,2\Sigma_{\ell,i,j})} [\phi(u)\phi(v)] \right|$$

$$= 2\left| \|h_{\ell-1,i}\|_2\|h_{\ell-1,j}\|_2 F(\theta_{\ell,i,j}) - \sqrt{(\mathbf{K}_{\ell-1})_{i,i}(\mathbf{K}_{\ell-1})_{j,j}} F(\tau_{\ell,i,j}) \right|$$

$$\le \widetilde{O}(m^{-1/2}).$$

That is,

$$\|\widehat{\mathbf{K}}_\ell - \mathbf{K}_\ell\|_\infty \le \widetilde{O}(m^{-1/2}). \tag{7}$$

Combining Eqs. (5) and (7) together, we get that

$$\max_{(i,j)\in[n]\times[n]} |h_{\ell,i}^\top h_{\ell,j} - (\mathbf{K}_\ell)_{i,j}| \leq \widetilde{O}(m^{-1/2})$$

holds with probability at least $1 - \frac{\delta}{L}$ for $\ell = h + 1$.

By induction, we have proved that for the first $L - 1$ layers, the intermediate correlation $h_{\ell,i}^\top h_{\ell,j}$ is close to the intermediate Gram matrix $(\mathbf{K}_\ell)_{i,j}$. Now, we consider the last layer. Recall $G_L$ is defined by Eq. (4), which has the same form as the correlation matrix of a two-layer over-parameterized neural network with input data $\{h_{L-1,i}\}_{i\in[n]}$. Define

$$(\widehat{\mathbf{K}}_L)_{i,j} := h_{L-1,i}^\top h_{L-1,j} \cdot \mathop{\mathbb{E}}_{w\sim\mathcal{N}(0,2I_m)} \left[\phi'(w^\top h_{L-1,i})\phi'(w^\top h_{L-1,j})\right].$$

Then, by the analysis of the two-layer case (see for example Song & Yang (2019); Du et al. (2019b)), we have

$$\|G_L - \widehat{\mathbf{K}}_L\| \leq \frac{\lambda_L}{8},$$

if $m = \Omega(\lambda_L^{-2} n^2 \log(n/\delta))$, where $\lambda_L := \lambda_{\min}(\mathbf{K}_L)$. It remains to bound $\|\widehat{\mathbf{K}}_L - \mathbf{K}_L\|_\infty$. Equivalently, for any $(i,j) \in [n] \times [n]$,

$$\max_{(i,j)\in[n]\times[n]} \left| \mathop{\mathbb{E}}_{(u,v)\sim\mathcal{N}(0,2\widehat{\Sigma}_{L,i,j})} [\phi'(u)\phi'(v)] - \mathop{\mathbb{E}}_{(u,v)\sim\mathcal{N}(0,2\Sigma_{L,i,j})} [\phi'(u)\phi'(v)] \right|.$$

The expectation has the following analytic form:

$$\mathop{\mathbb{E}}_{(z_1,z_2)\sim\mathcal{N}(0,\Sigma)} [\phi'(z_1)\phi'(z_2)] = \frac{1}{4} + \frac{\sin^{-1}(\rho)}{2\pi} \quad \text{with} \quad \Sigma = \begin{bmatrix} p^2 & \rho pq \\ \rho pq & q^2 \end{bmatrix}.$$

By the analysis of the $(L-1)$-layer, we know that $|\rho_{L,i,j} - \widehat{\rho}_{L,i,j}| \leq \widetilde{O}(m^{-1/2})$, where $\rho_{L,i,j} := \cos(\tau_{L,i,j})$ and $\widehat{\rho}_{L,i,j} := \cos(\theta_{L,i,j})$. Also, notice that $\cos(\tau_{L,i,j}) = F(\tau_{L-1,i,j}) \in [0, 1/2]$ by Eq. (6). Hence, the derivative of the expectation is bounded, and by Taylor's theorem, we have

$$\|\widehat{\mathbf{K}}_L - \mathbf{K}_L\|_\infty \leq \widetilde{O}(m^{-1/2}).$$

It implies that $\|\widehat{\mathbf{K}}_L - \mathbf{K}_L\| \leq \frac{\lambda_L}{8}$, which further implies that

$$\|G_L - \mathbf{K}_L\| \leq \frac{\lambda_L}{4}.$$

Equivalently, we get that

$$\lambda_{\min}(G_L) \geq \frac{3}{4}\lambda_L$$

with probability at least $1 - \delta$.

The lemma is then proved. $\qquad\square$

## E.2 Bounds on the Least Eigenvalue During Optimization

In this section, we generalize the Lemma C.5 in Brand et al. (2021) into multiple layer neural network

**Lemma E.3** (Bounds on the least eigenvalue, multiple layer neural network version of Lemma C.5 of Brand et al. (2021))**.** *Suppose $m = \Omega(\lambda_\ell^{-2} n^2 \log(n/\delta))$, with probability least $1 - \delta$, for any set of weights $W_1, \cdots W_L$ satisfying*

$$\|W_\ell - W_\ell(0)\| \leq R.$$

*then the following holds*

$$\|G_L(W) - G_L(W(0))\|_F \leq \lambda_L/2.$$

*Proof.* Recall $(G_L)(W)_{i,j} = \frac{1}{m}\sum_{r=1}^m \phi'(\langle(W_L)_r, h_{i,L-1}\rangle)\phi'(\langle(W_L)_r, h_{j,L-1}\rangle)h_{i,L-1}^\top h_{j,L-1}$. For simplicity, let $z_{i,r} := (W_L)_r^\top h_{i,L-1}$ and $z_{i,r}(0) := (W_L(0))_r^\top h_{i,L-1}(0)$.

$$|(G_L(W))_{i,j} - (G_L(W(0)))_{i,j}|$$

$$= \left| h_{i,L-1}^\top h_{j,L-1} \frac{1}{m}\sum_{r=1}^m \phi'(z_{i,r})\phi'(z_{j,r}) - h_{i,L-1}(0)^\top h_{j,L-1}(0)\frac{1}{m}\sum_{r=1}^m \phi'(z_{i,r}(0))\phi'(z_{j,r}(0)) \right|$$

$$\leq \left| h_{i,L-1}^\top h_{j,L-1} - h_{i,L-1}(0)^\top h_{j,L-1}(0) \right| \cdot \frac{1}{m}\sum_{r=1}^m \phi'(z_{i,r})\phi'(z_{j,r})$$

$$+ \left| h_{i,L-1}(0)^\top h_{j,L-1}(0) \right| \cdot \frac{1}{m}\left| \sum_{r=1}^m \phi'(z_{i,r})\phi'(z_{j,r}) - \phi'(z_{i,r}(0))\phi'(z_{j,r}(0)) \right|$$

$$\leq \left| h_{i,L-1}^\top h_{j,L-1} - h_{i,L-1}(0)^\top h_{j,L-1}(0) \right| + \frac{\lambda_L + \widetilde{O}(m^{-1/2})}{m}\left| \sum_{r=1}^m \phi'(z_{i,r})\phi'(z_{j,r}) - \phi'(z_{i,r}(0))\phi'(z_{j,r}(0)) \right|,$$

where the last step follows from $\phi'(x) \in \{0,1\}$ and Lemma E.2.

To upper bound the above two terms, we first need to bound the move of $h_{i,L-1}$. For any $\ell \in [L-1]$, we have

$$\begin{aligned}
\|h_{i,\ell} - h_{i,\ell}(0)\|_2 &= \|\phi(W_\ell h_{i,\ell-1}) - \phi(W_\ell(0)h_{i,\ell-1}(0))\|_2 \\
&\leq \|\phi(W_\ell h_{i,\ell-1}) - \phi(W_\ell h_{i,\ell-1}(0))\|_2 \\
&\quad + \|\phi(W_\ell h_{i,\ell-1}(0)) - \phi(W_\ell(0)h_{i,\ell-1}(0))\|_2 \\
&\leq (\|W_\ell - W_\ell(0)\| + \|W_\ell(0)\|) \cdot \|h_{i,\ell-1} - h_{i,\ell-1}(0)\|_2 \\
&\quad + \|W_\ell - W_\ell(0)\| \cdot \|h_{i,\ell-1}(0)\|_2 \\
&\leq (R + c_W)\|h_{i,\ell-1} - h_{i,\ell-1}(0)\|_2 + R(1+\epsilon),
\end{aligned}$$

where $c_W := \|W_\ell(0)\| \leq 3$ by the well-known deviations bounds concerning the singular values of Gaussian random matrices (Rudelson & Vershynin (2010)). Also, when $\ell = 0$, we have $\|h_{i,0} - h_{i,0}(0)\|_2 = 0$. Hence, we get that for all $\ell \in [L-1]$, $i \in [n]$,

$$\|h_{i,\ell} - h_{i,\ell}(0)\|_2 \leq \sqrt{1+\epsilon}R(2c_W)^\ell,$$

since $R \ll 1$ by our choice of $R$.

Hence, it implies that

$$\begin{aligned}
&\left| h_{i,L-1}^\top h_{j,L-1} - h_{i,L-1}(0)^\top h_{j,L-1}(0) \right| \\
&\leq \left| (h_{i,L-1} - h_{i,L-1}(0))^\top h_{j,L-1} \right| + \left| h_{i,L-1}(0)^\top (h_{j,L-1} - h_{j,L-1}(0)) \right| \\
&\leq 2(1+\epsilon)R(2c_W)^{L-1}.
\end{aligned}$$

Similarly,

$$\frac{\lambda_L + \widetilde{O}(m^{-1/2})}{m}\left| \sum_{r=1}^m \phi'(z_{i,r})\phi'(z_{j,r}) - \phi'(z_{i,r}(0))\phi'(z_{j,r}(0)) \right|$$

$$= \frac{\lambda_L + \widetilde{O}(m^{-1/2})}{m}\left| \mathbf{1}_{(W_L)_r^\top h_{i,L-1}>0,(W_L)_r^\top h_{j,L-1}>0} - \mathbf{1}_{(W_L(0))_r^\top h_{i,L-1}(0)>0,(W_L(0))_r^\top h_{j,L-1}(0)>0} \right|.$$

By our assumption, we have $\|(W_L)_r - (W_L(0))_r\|_2 \leq R$. We also know that $\|h_{i,L-1} - h_{i,L-1}(0)\|_2 \leq \sqrt{1+\epsilon}R(2c_W)^{L-1}$. Then, we can follow the proof in Du et al. (2019b); Song & Yang (2019) and define the event $A_{i,r} := \exists w, h : \|w - (W_L(0))_r\|_2 \leq R, \|h - h_{i,L-1}(0)\|_2 \leq O(R)$ such that $\mathbf{1}_{w^\top h>0} \neq \mathbf{1}_{(W_L(0))_r^\top h_{i,L-1}(0)>0}$. We have $A_{i,r}$ happens if and only if $\left| ((W_L(0))_r + \Delta w)^\top h_{i,L-1}(0) \right| < O(R(R + m^{-1/2}))$ for some fixed vector $\Delta w$ of length at most $R$. It implies that

$$\Pr[A_{i,r}] = \Pr_{z \sim \mathcal{N}(0,1)}[|z| < R + o(R)] \leq O(R).$$

Hence, using the same proof in the previous work, we get that

$$\frac{\lambda_L + \widetilde{O}(m^{-1/2})}{m} \left| \sum_{r=1}^{m} \phi'(z_{i,r})\phi'(z_{j,r}) - \phi'(z_{i,r}(0))\phi'(z_{j,r}(0)) \right| \leq (\lambda_L + m^{-1/2})R.$$

Putting them together, we have

$$|(G_L(W))_{i,j} - (G_L(W(0)))_{i,j}| \leq (2(1+\epsilon)(2c_W)^{L-1} + \lambda_L)R.$$

And by our choice of $R$, we get that

$$\|G_L(W) - G_L(W(0))\|_F \leq (2(1+\epsilon)(2c_W)^{L-1} + \lambda_L)nR \leq \lambda_L/2.$$

The lemma is then proved. $\qquad\square$

## F  CONVERGENCE ANALYSIS OF ALGORITHM 2

In this section, we analyze the convergence behavior of Algorithm 2.

### F.1  PRELIMINARY

We recall the initialization of our neural network.

**Definition F.1** (Initialization). *Let $m = m_\ell$ for all $\ell \in [L]$. Let $m_0 = d$. We assume weights are initialized as*

- *Each entry of weight vector $a \in \mathbb{R}^{m_L}$ is i.i.d. sampled from $\{-\frac{1}{\sqrt{m_L}}, +\frac{1}{\sqrt{m_L}}\}$ uniformly at random.*

- *Each entry of weight matrices $W_\ell \in \mathbb{R}^{m_\ell \times m_{\ell-1}}$ sampled from $\mathcal{N}(0, 2/m_\ell)$.*

We also restate the architecture of our neural network here.

**Definition F.2** (Architecture). *Our neural network is a standard $L$-layer feed-forward neural network, with the activation functions defined as a scaled version of shifted ReLU activation: $\phi(x) = \sqrt{c_b}\mathbf{1}[x > \sqrt{2/m}b]x$, where $c_b := (2(1 - \Phi(b) + b\phi(b)))^{-1/2}$. Here $b$ is a threshold value we will pick later. At last layer, we use a scaled version of a vector with its entry being Rademacher random variables. We define the neural network function $f : \mathbb{R}^{m_0} \to \mathbb{R}$ as*

$$f(W, x_i) = a^\top \phi(W_L \phi(W_{L-1}\phi(\ldots \phi(W_1 x_i)))).$$

*We measure the loss of the neural network via squared-loss function:*

$$\mathcal{L}(W) = \frac{1}{2}\sum_{i=1}^{n}(f(x_i) - y_i)^2.$$

*We use $f_t : \mathbb{R}^{m_0 \times n} \to \mathbb{R}^n$ denote the prediction of our network:*

$$f_t(X) = [f(W(t), x_1), \ldots, f(W(t), x_n)]^\top.$$

We state two assumptions here.

**Assumption F.3** (Small Spectral Norm). *Let $t \in \{0, \ldots, T\}$ and let $R \leq 1$ be a parameter. We assume*

$$\max_{\ell \in [L]} \|W_\ell(t) - W_\ell(0)\| \leq R.$$

*Later, we will invoke this assumption by specifying the choice of $R$.*

**Assumption F.4** (Sparsity). *Let $t \in \{0, \ldots, T\}$ and let $s \geq 1$ be an integer parameter. We assume*

$$\|\Delta D_{i,\ell}\|_0 \leq s, \forall \ell \in [L], i \in [n].$$

*Later, we will invoke this assumption by specifying the choice of $s$.*

Finally, throughout this entire section, we will assume $m_\ell = m$ for any $\ell \in [L]$.

## F.2 TECHNICAL LEMMAS

We first show that during initialization, by using our threshold ReLU activation, the vector $h_{i,\ell}$ is sparse, hence the diagonal matrix $D_{i,\ell}$ is sparse as well.

**Lemma F.5** (Sparse initialization). *Let $\sigma_b(x) = \max\{x - b, 0\}$ be the threshold ReLU activation with threshold $b > 0$. After initialization, with probability $1 - nL \cdot e^{-\Omega(me^{-b^2m/4})}$, it holds for all $i \in [n]$ and $\ell \in [L]$, $\|h_{i,\ell}\|_0 \le O(m \cdot e^{-b^2m/4})$.*

*Proof.* We fix $i \in [n]$ and $\ell \in [L]$, since we will union bound over all $i$ and $\ell$ at last. Let $u_i \in \mathbb{R}^m$ be a fixed vector and $W_{\ell,r}$ to denote the $r$-th row of $W_\ell$, then by the concentration of Gaussian, we have

$$\Pr[\sigma_b(\langle W_{\ell,r}, u_i\rangle) > 0] = \Pr_{z \sim \mathcal{N}(0, \frac{2}{m})}[z > b] \le \exp(-b^2m/4).$$

Let $S$ be the following index set $S := \{r \in [m] : \langle W_{\ell,r}, u_i\rangle > b\}$, the above reasoning means that for the indicator random variable $\mathbf{1}[r \in S]$, we have

$$\mathbb{E}[\mathbf{1}[r \in S]] \le \exp(-b^2m/4).$$

Use Bernstein's inequality (Lemma A.3) we have that for all $t > 0$,

$$\Pr[|S| > k + t] \le \exp(-\frac{t^2/2}{k + t/3}),$$

where $k := m \cdot \exp(-b^2m/4)$. By picking $t = k$, we have

$$\Pr[|S| > 2k] \le \exp(\frac{-3k}{8}).$$

Note that $|S|$ is essentially the quantity $\|h_{i,\ell}\|_0$, hence we can union bound over all $\ell$ and $i$ and with probability at least

$$1 - nL \cdot \exp(-\Omega(m \cdot \exp(b^2m/4))),$$

we have $\|h_{i,\ell}\|_0 \le 2m \cdot \exp(-b^2m/4)$. $\qquad\qquad\square$

**Remark F.6.** *The above lemma shows that by using the shifted ReLU activation, we make sure that all $h_{i,\ell}$ are sparse after initialization. Specifically, we use $k := m \cdot \exp(-b^2m/4)$ as a sparsity parameter. Later, we might re-scale $b$ so that the probability becomes $\exp(-b^2/2)$. We stress that such re-scaling does not affect the sparsity of our initial vectors. If we re-scale $b$ and choose it as $\sqrt{0.4\log m}$, then $k = m^{0.8}$ and hence with high probability, $\|h_{i,\ell}\|_0 \le O(m^{0.8})$.*

*As a direct consequence, we note that all initial $D_{i,\ell}$ are $k$-sparse as well.*

We state a lemma that handles the $\ell_2$ norm of $h_{i,\ell}$ when one uses *truncated Gaussian distribution* instead. Due to the length and the delicacy of the proof, we defer it to Section G.

**Lemma F.7** (Restatement of Lemma G.6). *Let $b > 0$ be a fixed scalar. Let the activation function $\phi(x) := \sqrt{c_b}\mathbf{1}[x > \sqrt{2/m}b]x$, where $c_b := (2(1 - \Phi(b) + b\phi(b)))^{-1/2}$. Let $\epsilon \in (0, 1)$, then over the randomness of $W(0)$, with probability at least $1 - O(nL) \cdot \exp(-\Omega(m\exp(-b^2/2)\epsilon^2/L^2))$, we have*

$$\|h_{i,\ell}\|_2 \in [1 - \epsilon, 1 + \epsilon], \forall i \in [n], \ell \in [L].$$

We remark that the parameter $e^{-b^2/2}m$ captures the *sparsity* during the initialization.

The second lemma handles the consecutive product that appears naturally in the gradient computation.

**Lemma F.8** (Variant of Lemma 7.3 in Allen-Zhu et al. (2019a)). *Suppose $m \ge \Omega(nL\log(nL))$. With probability at least $1 - e^{-\Omega(k/L^2)}$ over the randomness of initializations $W_1(0), \ldots, W_L(0) \in \mathbb{R}^{m \times m}$, for all $i \in [n]$ and $1 \le a \le b \le L$,*

(a) $\|W_b D_{i,b-1} W_{b-1} \ldots D_{i,a} W_a\| \leq O(\sqrt{L})$.

(b) $\|W_b D_{i,b-1} W_{b-1} \ldots D_{i,a} W_a v\|_2 \leq 2\|v\|_2$, for all vectors with $\|v\|_0 \leq O(\frac{m}{L \log m})$.

(c) $\|u^\top W_b D_{i,b-1} W_{b-1} \ldots D_{i,a} W_a\|_2 \leq O(1)\|u\|_2$, for all vectors $u$ with $\|u\|_0 \leq O(\frac{m}{L \log m})$.

(d) For any intergers $1 \leq s \leq O(\frac{m}{L \log m})$, with probability at least $1 - e^{-\Omega(s \log m)}$ over the randomness of initializations $W_1(0), \ldots, W_L(0) \in \mathbb{R}^{m \times m}$, $|u^\top W_b D_{i,b-1} W_{b-1} \ldots D_{i,a} W_a v| \leq \|u\|_2 \|v\|_2 \cdot O(\frac{m}{L \log m})$ for all vectors $u, v$ with $\|u\|_0, \|v\|_0 \leq s$.

The proof is similarly to the original proof of the corresponding lemma in Allen-Zhu et al. (2019a), however we replace the bound on $h_{i,\ell}$ with our Lemma F.7. We highlight this does not change the bound, merely in expense of a worse probability.

The next lemma concerns the bound on the product being used in backpropagation.

**Lemma F.9** (Variant of Lemma 7.4 in Allen-Zhu et al. (2019a)). *Suppose $m \geq \Omega(nL \log(nL))$. Then with probability at least $1 - e^{-\Omega(\log^2 m)}$, for all $i \in [n]$, $\ell \in [L]$,*

$$\|a^\top D_{i,L} W_L \ldots D_{i,\ell} W_\ell\|_2 \leq O(\frac{\log m}{\sqrt{m}} \sqrt{L}).$$

*Proof.* By Lemma F.8 part (a), we know that $\|W_L \ldots D_{i,\ell} W_\ell\| \leq O(\sqrt{L})$, consequently, $\|D_{i,L} W_L \ldots D_{i,\ell} W_\ell\| \leq O(\sqrt{L})$ since $\|D_{i,L}\| \leq 1$. This means for any vector $u \in \mathbb{R}^m$, with probability at least $1 - e^{-\Omega(k/L^2)}$, we have

$$\|D_{i,L} W_L \ldots D_{i,\ell} W_\ell u\|_2 \leq O(\sqrt{L})\|u\|_2.$$

Conditioning on this event and using the randomness of Rademacher vector $a$, we have

$$\Pr[|a^\top D_{i,L} W_L \ldots D_{i,\ell} W_\ell u| \leq t \cdot \sqrt{L}\|u\|_2] \geq 1 - 2\exp(-\frac{mt^2}{2}),$$

pick $t = \frac{\log m}{\sqrt{m}}$, we know that

$$\Pr[|a^\top D_{i,L} W_L \ldots D_{i,\ell} W_\ell u| \leq \frac{\log m}{\sqrt{m}} \cdot \sqrt{L}\|u\|_2] \geq 1 - 2\exp(-\frac{\log^2 m}{2}).$$

This means with probability at least $1 - e^{-\Omega(\log^2 m)}$, we have

$$\|a^\top D_{i,L} W_L \ldots D_{i,\ell} W_\ell\|_2 \leq O(\frac{\log m}{\sqrt{m}} \sqrt{L}).$$

$\square$

The next several lemmas bound norms after small perturbation.

**Lemma F.10** (Lemma 8.2 in Allen-Zhu et al. (2019a)). *Suppose Assumption F.3 is satisfied with $R \leq O(\frac{1}{L^{9/2} \log^3 m})$. With probability at least $1 - e^{-\Omega(mR^{2/3}L)}$,*

(a) $\Delta g_{i,\ell}$ *can be written as* $\Delta g_{i,\ell} = \Delta g_{i,\ell,1} + \Delta g_{i,\ell,2}$ *where* $\|\Delta g_{i,\ell,1}\|_2 \leq O(RL^{3/2})$ *and* $\|\Delta g_{i,\ell,2}\|_\infty \leq O(\frac{RL^{5/2}\sqrt{\log m}}{\sqrt{m}})$.

(b) $\|\Delta D_{i,\ell}\|_0 \leq O(mR^{2/3}L)$ *and* $\|(\Delta D_{i,\ell})g_{i,\ell}\|_2 \leq O(RL^{3/2})$.

(c) $\|\Delta g_{i,\ell}\|_2, \|\Delta h_{i,\ell}\|_2 \leq O(RL^{5/2}\sqrt{\log m})$.

**Remark F.11.** *Lemma F.10 establishes the connection between parameter $R$ and $s$ of Assumption F.3 and F.4. As long as $R$ is small, then we have $s = O(mR^{2/3}L)$. Such a relation enables us to pick $R$ to our advantage and ensure the sparsity of $\Delta D_{i,\ell}$ is sublinear in $m$, and hence the update time per iteration is subquadratic in $m$.*

**Lemma F.12** (Lemma 8.6 in Allen-Zhu et al. (2019a))**.** *Suppose Assumption F.4 is satisfied with* $1 \leq s \leq O(\frac{m}{L^3 \log m})$ *and* $m \geq \Omega(nL \log(nL))$, *with probability at least* $1 - e^{-\Omega(s \log m)}$ *over the randomness of* $W(0)$, *for every* $i \in [n]$, $1 \leq a \leq b \leq L$, *we have*

*(a)* $\|W_b(0)(D_{i,b-1}(0) + \Delta D_{i,b-1}) \ldots (D_{i,a}(0) + \Delta D_{i,a})W_a(0)\| \leq O(\sqrt{L})$.

*(b)* $\|(W_b(0) + \Delta W_b)(D_{i,b-1}(0) + \Delta D_{i,b-1}) \ldots (D_{i,a}(0) + \Delta D_{i,a})(W_a(0) + \Delta W_a)\| \leq O(\sqrt{L})$ *if Assumption F.3 is satisfied with* $R \leq O(\frac{1}{L^{1.5}})$.

**Corollary F.13.** *Suppose Assumption F.3 is satisfied with* $R \leq O(\frac{1}{L^{4.5} \log^3 m})$ *and* $m \geq \Omega(nL \log(nL))$, *with probability at least* $1 - e^{-\Omega(\log^2 m)}$ *over the randomness of* $W(0)$ *and* $a$, *for any* $i \in [n]$ *and* $\ell \in [L]$, *we have*

$$\|a^\top (D_{i,L}(0) + \Delta D_{i,L}(0))(W_L(0) + \Delta W_L) \ldots (W_{\ell+1}(0) + \Delta W_{\ell+1})(D_{i,\ell}(0) + \Delta D_{i,\ell})\|_2$$
$$\leq \widetilde{O}(\sqrt{L/m}).$$

*Proof.* We first note that by Lemma F.10 and the choice of $R$, we know that Assumption F.4 is satisfied with $1 \leq s \leq O(\frac{m}{L^2 \log^2 m})$.

The proceeding proof is identical to Lemma F.9, use $P$ to denote the product $(D_{i,L}(0) + \Delta D_{i,L}(0))(W_L(0) + \Delta W_L) \ldots (W_{\ell+1}(0) + \Delta W_{\ell+1})(D_{i,\ell}(0) + \Delta D_{i,\ell})$. Per Lemma F.12, we know that with the conditions stated, we have

$$\|P\| \leq O(\sqrt{L}),$$

conditioning on this even, for any vector $u \in \mathbb{R}^m$, with probability at least $1 - e^{-\Omega(s \log m)}$, we know that

$$\|Pu\|_2 \leq O(\sqrt{L})\|u\|_2.$$

Use the randomness of vector $a$, we have

$$\Pr[|a^\top Pu| \leq t \cdot \sqrt{L}\|u\|_2] \geq 1 - 2\exp(-\frac{mt^2}{2}).$$

Pick $t = \frac{\log m}{\sqrt{m}}$, we are done. $\qquad \square$

**Lemma F.14** (Variant of Lemma 8.7 in Allen-Zhu et al. (2019a))**.** *Suppose Assumption F.3 is satisfied with* $R \leq O(\frac{1}{L^{4.5} \log^3 m})$ *and* $m \geq \Omega(nL \log(nL))$, *with probability at least* $1 - e^{-\Omega(\log^2 m)}$ *over the randomness of* $W(0)$ *and* $a$, *for all* $i \in [n]$, $\ell \in [L]$. *Define*

$$u := a^\top (D_{i,L}(0) + \Delta D_{i,L})(W_L(0) + \Delta W_L(0)) \ldots (W_{\ell+1}(0) + \Delta W_{\ell+1})(D_{i,\ell}(0) + \Delta D_{i,\ell}),$$
$$u_0 := a^\top D_{i,L}(0)W_L(0) \ldots W_{\ell+1}(0)D_{i,\ell}(0).$$

*It satisfies that*

$$\|u - u_0\|_2 \leq \widetilde{O}(\sqrt{L/m}).$$

*Proof.* Note that we can invoke Corollary F.13 to give an upper bound on $\|u\|_2$ with probability $1 - e^{-\Omega(\log^2 m)}$,

$$\|u\|_2 \leq \widetilde{O}(\sqrt{L/m}).$$

Similarly, we can use Lemma F.9 to give an upper bound on $\|u_0\|_2$ with probability $1 - e^{-\Omega(\log^2 m)}$,

$$\|u_0\|_2 \leq \widetilde{O}(\sqrt{L/m}).$$

Finally, by triangle inequality, we get

$$\|u - u_0\|_2 \leq \widetilde{O}(\sqrt{L/m}),$$

with the desired probability. $\qquad \square$

### F.3 BOUNDS ON INITIALIZATION

In the following lemma, we generalize the Lemma C.2 in Brand et al. (2021) into multiple layer neural networks.

**Lemma F.15** (Bounds on initialization, multiple layer version of Lemma C.2 in Brand et al. (2021)). *Suppose $m = \Omega(nL \log(nL))$, then with probability $1 - O(nL) \cdot e^{-\Omega(k/L^2)}$, we have the following*

- $f(W, x_i) = O(1)$, $\forall i \in [n]$.

- *With probability at least $1 - e^{-\Omega(\log^2 m)}$, we have $\|J_{\ell,0,i}\| = \widetilde{O}(\sqrt{L/m})$, $\forall \ell \in [L]$, $\forall i \in [n]$.*

**Remark F.16.** *The bound of $m$ in Brand et al. (2021) has a linear dependence on $n$ because they need to bound $\|W(0)\|_2$. If we do not need to bound $\|W(0)\|_2$, then $m$ does not depend linearly on $n$.*

*Proof.* We will prove the two parts of the statement separately.

**Part 1:** By definition, for any $i \in [n]$, we have

$$f(W, x_i) = a^\top \phi(W_L(\phi(\cdots \phi(W_1 x_i)))).$$

We shall make use Lemma F.7 here: with probability at least $1 - O(nL) \cdot \exp(-\Omega(k/L^2))$, we have $\|h_{i,L}\|_2 \in [0.9, 1.1]$. Recall that $a \in \mathbb{R}^m$ has each of its entry being a Rademacher random variable scaled by $1/m$, hence it's $1/m$-subgaussian. Using the concentration of subgaussian (Lemma A.5), we have that

$$\Pr[|f(W, x_i)| \geq t] \leq 2 \exp(-\frac{t^2 m}{2\|h_{i,L}\|_2}),$$

finally by noticing that $t = \Theta(1)$ and $\|h_{i,L}\|_2 \in [0.9, 1.1]$, we conclude that

$$\Pr[|f(W, x_i)| \geq O(1)] \leq 2 \exp(-\Omega(m)),$$

union bounding over all $i$, we conclude that

$$\Pr[f(W, x_i) \geq O(1)] \leq O(n) \cdot \exp(-\Omega(m)), \forall i \in [n].$$

**Part 2:** We will combine Lemma F.7 and F.9, with probability at least $1 - O(nL) \cdot e^{-\Omega(k/L^2)}$, we have $\|h_{i,\ell}\|_2 \in [0.9, 1.1]$ and with probability at least $1 - e^{-\Omega(\log^2 m)}$, we have $\|a^\top D_{i,L} \ldots W_\ell\|_2 \leq \widetilde{O}(\sqrt{L/m})$. Combine them together, we know that with probability at least $1 - e^{-\Omega(\log^2 m)}$, $\|J_{\ell,0,i}\| \leq \widetilde{O}(\sqrt{L/m})$. □

**Remark F.17.** *By utilizing the structure of vector $a$, we can show that with a weaker probability $(1 - e^{-\Omega(\log^2 m)})$, $f(W, x_i)$ has even smaller magnitude ($\widetilde{O}(\frac{1}{\sqrt{m}})$). For our purpose, $f(W, x_i) = O(1)$ suffices.*

### F.4 BOUNDS ON SMALL PERTURBATION

In the following, we generalize the Lemma C.4 in Brand et al. (2021) into multiple layer neural network. For the simplicity of notation, we set all $m_\ell$ to be $m$.

**Lemma F.18** (multiple layer version of Lemma C.4 in Brand et al. (2021)). *Suppose $R \leq O(\frac{1}{L^{4.5} \log^3 m})$ and $m = \Omega(nL \log(nL))$. With probability at least $1 - e^{-\Omega(\log^2 m)}$ over the random initialization of $W(0) = \{W_1(0), W_2(0), \cdots W_L(0)\}$, the following holds for any set of weights $W_1, \cdots, W_L$ satisfying*

$$\|W_\ell - W_\ell(0)\| \leq R, \forall \ell \in [L]$$

- $\|J_{W_\ell, x_i} - J_{W_\ell(0), x_i}\|_2 = \widetilde{O}(\sqrt{L/m})$.

- $\|J_{W_\ell} - J_{W_\ell(0)}\|_F = \widetilde{O}(n^{1/2}\sqrt{L/m})$.

- $\|J_{W_\ell}\|_F = \widetilde{O}(n^{1/2}\sqrt{L/m})$.

*Proof.* **Part 1.** To simplify the notation, we ignore the subscripts $i$ below.

Consider the following computation:

$$
\begin{aligned}
\|J_{W_\ell, x_i} - J_{W_\ell(0), x_i}\|_2 &= \|uh_{\ell-1}^\top - u_0 h_{\ell-1}^\top(0)\|_F \\
&= \|u(h_{\ell-1}(0) + \Delta h_{\ell-1})^\top - u_0 h_{\ell-1}^\top(0)\|_F \\
&= \|(u - u_0)h_{\ell-1}^\top(0) + u(\Delta h_{\ell-1}^\top)\|_F \\
&\le \|(u - u_0)h_{\ell-1}^\top(0)\|_F + \|u(\Delta h_{\ell-1}^\top)\|_F \\
&\le \|u - u_0\|_2 \cdot \|h_{\ell-1}(0)\|_2 + \|u\|_2 \cdot \|\Delta h_{\ell-1}\|_2.
\end{aligned}
$$

where

$$
u := (D_\ell(0) + \Delta D_\ell)\left(\prod_{k=\ell+1}^{L}(W_k(0) + \Delta W_k)^\top(D_k(0) + \Delta D_k)\right)a
$$

$$
u_0 := D_\ell(0)\left(\prod_{k=\ell+1}^{L}W_k(0)^\top D_k(0)\right)a
$$

By Lemma F.14, we know that with probability at least $1 - e^{-\Omega(\log^2 m)}$, we have

$$
\|u - u_0\|_2 \le \widetilde{O}(\sqrt{L/m}).
$$

By Lemma F.7, we know that with probability at least $1 - O(nL) \cdot e^{-k/L^2}$, we have

$$
\|h_{\ell-1}(0)\|_2 \in [0.9, 1.1].
$$

By Corollary F.13, we know that with probability at least $1 - e^{-\Omega(\log^2 m)}$, we have

$$
\|u\|_2 \le \widetilde{O}(\sqrt{L/m}).
$$

By Lemma F.10, we know that with probability at least $1 - e^{-\Omega(mR^{2/3}L)}$, we have

$$
\|\Delta h_{\ell-1}\|_2 \le \widetilde{O}(RL^{2.5}).
$$

Note that due to the choice of $R$, we know that

$$
\|\Delta h_{\ell-1}\|_2 \le 1.
$$

Taking a union bound over all events, with probability at least $1 - e^{-\Omega(\log^2 m)}$, we achieve the following bound:

$$
\|J_{W_\ell, x_i} - J_{W_\ell(0), x_i}\|_2 \le \widetilde{O}(\sqrt{L/m}).
$$

**Part 2.** Note that the squared Frobenious norm is just the sum of all squared $\ell_2$ norm of rows, hence

$$
\|J_{W_\ell} - J_{W_\ell(0)}\|_F \le \widetilde{O}(n^{1/2}\sqrt{L/m}).
$$

**Part 3.** We will prove by triangle inequality:

$$
\begin{aligned}
\|J_{W_\ell}\|_F &\le \|J_{W_\ell(0)}\|_F + \|J_{W_\ell} - J_{W_\ell(0)}\|_F \\
&\le \widetilde{O}(n^{1/2}\sqrt{L/m}) + \widetilde{O}(n^{1/2}\sqrt{L/m}) \\
&= \widetilde{O}(n^{1/2}\sqrt{L/m}).
\end{aligned}
$$

$\square$

### F.5 PUTTING IT ALL TOGETHER

In this section, we will prove the following core theorem that analyzes the convergence behavior of Algorithm 2:

**Theorem F.19** (Formal version of Theorem 1.1). *Suppose the width of the neural network satisfies $m = \Omega(\lambda_L^{-2} n^2 L^2)$, then with probability at least $1 - e^{-\Omega(\log^2 m)}$ over the randomness of the initialization of the neural network and the randomness of the algorithm, Algorithm 2 satisfies*

$$\|f_{t+1} - y\|_2 \leq \frac{1}{2}\|f_t - y\|_2.$$

Before moving on, we introduce several definitions and prove some useful facts related to them.

**Definition F.20** (function J). *We define*

$$\mathsf{J}_\ell(Z_1, \ldots, Z_L)_i := D_{i,\ell}(Z_\ell) \prod_{k=\ell+1}^{L} Z_k^\top D_{i,k}(Z_k) a(h_i(Z_1, \ldots, Z_{\ell-1}))^\top \quad \in \mathbb{R}^{m_\ell \times m_{\ell-1}}$$

*where*

$$D_{i,\ell}(Z_\ell) := \mathrm{diag}(\phi'(Z_\ell h_i(Z_1, \ldots, Z_{\ell-1}))), \qquad \in \mathbb{R}^{m_\ell \times m_\ell}$$
$$h_i(Z_1, \ldots, Z_{\ell-1}) := \phi(Z_{\ell-1}(\phi(Z_{\ell-2} \cdots (\phi(Z_1 x_i))))) \qquad \in \mathbb{R}^{m_{\ell-1}}$$

**Fact F.21.** *Let $\mathsf{J}_\ell$ denote the function be defined as Definition F.20. For any $t \in \{0, \ldots, T\}$, we have*

$$f_{t+1} - f_t = \sum_{\ell=1}^{L} \left( \int_0^1 \mathsf{J}_\ell((1-s)W(t) + sW(t+1))\mathrm{d}s \right)^\top \cdot \mathrm{vec}(W_\ell(t+1) - W_\ell(t)),$$

*Proof.* For $i \in [n]$, consider the $i$-th coordinate.

$$
\begin{aligned}
(f_{t+1} - f_t)_i &= \int_0^1 f((1-s)W(t) + sW(t+1), x_i)' \mathrm{d}s \\
&= \int_0^1 \sum_{\ell=1}^{L} \left( \frac{\partial f}{\partial W_\ell}((1-s)W(t) + sW(t+1), x_i) \right)^\top \cdot \mathrm{vec}(W_\ell(t+1) - W_\ell(t)) \mathrm{d}s \\
&= \sum_{\ell=1}^{L} \left( \int_0^1 \mathsf{J}_\ell((1-s)W(t) + sW(t+1))_i \mathrm{d}s \right)^\top \cdot \mathrm{vec}(W_\ell(t+1) - W_\ell(t)),
\end{aligned}
$$

Thus, we complete the proof. $\square$

**Fact F.22.** *For any $t \in \{0, \ldots, T\}$, we have $\mathsf{J}_\ell(W_1(t), \ldots, W_L(t)) = J_{\ell,t}$.*

*Proof.* In order to simplify the notation, we drop the term $t$ below.

We note that for $i \in [n]$, the $i$-th row of matrix $J_{\ell,t}$ is defined as

$$D_{i,\ell}(\prod_{k=\ell+1}^{L} W_k^\top D_{i,k})ah_{i,\ell-1}^\top,$$

where $D_{i,\ell} = \mathrm{diag}(\phi'(W_\ell h_{i,\ell-1}))$ and $h_{i,\ell-1} = \phi(W_{\ell-1}(\phi(W_{\ell-2} \ldots (\phi(W_1 x_i)))))$, this is essentially the same as $h_i(W_1, \ldots, W_{\ell-1})$ and $D_{i,\ell}(W_\ell)$. This completes the proof. $\square$

We state the range we require for parameter $R$:

**Definition F.23.** *We choose $R$ so that*

$$\frac{n}{\sqrt{mL}} \cdot \frac{1}{\lambda_L} \leq R \leq \frac{1}{L^{4.5} \log^3 m}.$$

**Remark F.24.** *Recall that the sparsity parameter $s$ is directly related to $R$: $s = O(mR^{2/3}L)$, hence to ensure the sparsity is small, we shall pick $R$ as small as possible. Specifically, if we pick $R$ to be $\frac{n}{\sqrt{mL}\lambda_L}$, then $s = O(\lambda_L^{-1}n\sqrt{mL})$, as long as $m \gg \lambda_L^{-2}n^2L$ then $s \approx O(\sqrt{m})$.*

Next, we pick the value of $m$:

**Definition F.25.** *We choose $m$ to be*
$$m \geq \Omega(\lambda_L^{-10/3}n^{10/3}L^{5/3}).$$

**Remark F.26.** *The choice of $m$ here makes sure that, as long as we pick $R$ matching its lower bound, then the sparsity $s = O(m^{0.8})$, this matches the other sparsity parameter $k$, which is also in the order of $O(m^{0.8})$.*

We use induction to prove the following two claims recursively.

**Definition F.27** (Induction hypothesis 1). *Let $t \in [T]$ be a fixed integer. We have*
$$\|W_\ell(t) - W_\ell(0)\| \leq R$$
*holds for any $\ell \in [L]$.*

**Definition F.28** (Induction Hypothesis 2). *Let $t \in [T]$ be a fixed integer. We have*
$$\|f_t - y\|_2 \leq \frac{1}{2}\|f_{t-1} - y\|_2.$$

Suppose the above two claims hold up to $t$, we prove they continue to hold for time $t + 1$. The second claim is more delicate, we are going to prove it first and we define
$$J_{\ell,t,t+1} := \int_0^1 \mathsf{J}_\ell\Big((1-s)W_t + sW_{t+1}\Big)\mathsf{d}s,$$
where $\mathsf{J}_\ell$ is defined as Definition F.20.

**Lemma F.29.** *Let $g_\ell^\star := (J_{\ell,t}J_{\ell,t}^\top)^{-1} \cdot \frac{1}{L}(f_t - y)$. We have*

$$\|f_{t+1} - y\|_2 \leq \|f_t - y - \sum_{\ell=1}^L J_{\ell,t}J_{\ell,t}^\top g_{\ell,t}\|_2$$

$$+ \sum_{\ell=1}^L \|(J_{\ell,t} - J_{\ell,t,t+1})J_{\ell,t}^\top g_\ell^\star\|_2$$

$$+ \sum_{\ell=1}^L \|(J_{\ell,t} - J_{\ell,t,t+1})J_{\ell,t}^\top(g_{\ell,t} - g_\ell^\star)\|_2. \tag{8}$$

*Proof.* Consider the following computation:
$$\|f_{t+1} - y\|_2$$
$$= \|f_t - y + (f_{t+1} - f_t)\|_2$$
$$= \|f_t - y + \sum_{\ell=1}^L J_{\ell,t,t+1} \cdot \text{vec}(W_{\ell,t+1} - W_{\ell,t})\|_2$$
$$= \|f_t - y - \sum_{\ell=1}^L J_{\ell,t,t+1} \cdot J_{\ell,t}^\top g_{\ell,t}\|_2$$
$$= \|f_t - y - \sum_{\ell=1}^L J_{\ell,t}J_{\ell,t}^\top g_{\ell,t} + \sum_{\ell=1}^L J_{\ell,t}J_{\ell,t}^\top g_{\ell,t} - \sum_{\ell=1}^L J_{\ell,t,t+1}J_{\ell,t}^\top g_{\ell,t}\|_2$$
$$\leq \|f_t - y - \sum_{\ell=1}^L J_{\ell,t}J_{\ell,t}^\top g_{\ell,t}\|_2 + \sum_{\ell=1}^L \|(J_{\ell,t} - J_{\ell,t,t+1})J_{\ell,t}^\top g_{\ell,t}\|_2$$
$$\leq \|f_t - y - \sum_{\ell=1}^L J_{\ell,t}J_{\ell,t}^\top g_{\ell,t}\|_2 + \sum_{\ell=1}^L \|(J_{\ell,t} - J_{\ell,t,t+1})J_{\ell,t}^\top g_\ell^\star\|_2 + \sum_{\ell=1}^L \|(J_{\ell,t} - J_{\ell,t,t+1})J_{\ell,t}^\top(g_{\ell,t} - g_\ell^\star)\|_2,$$
The second step follows from the definition of $J_{\ell,t,t+1}$ and simple calculus. $\qquad\square$

**Claim F.30** (1st term in Eq. (8)). *We have*

$$\|(f_t - y) - \sum_{\ell=1}^{L} J_{\ell,t} J_{\ell,t}^\top g_{\ell,t}\|_2 \leq \frac{1}{6} \|f_t - y\|_2.$$

*Proof.* We have

$$\|(f_t - y) - \sum_{\ell=1}^{L} J_{\ell,t} J_{\ell,t}^\top g_{\ell,t}\|_2 = \|\sum_{\ell=1}^{L} (\frac{1}{L}(f_t - y) - J_{\ell,t} J_{\ell,t}^\top g_{\ell,t})\|_2$$

$$\leq \sum_{\ell=1}^{L} \|\frac{1}{L} \cdot (f_t - y) - J_{\ell,t} J_{\ell,t}^\top g_{\ell,t}\|_2$$

$$\leq \frac{1}{6} \|f_t - y\|_2, \tag{9}$$

since $g_{\ell,t}$ is an $\epsilon_0$ ($\epsilon_0 \leq \frac{1}{6}$) approximate solution to regression problem

$$\min_g \|J_{\ell,t} J_{\ell,t}^\top g - \frac{1}{L}(f_t - y)\|_2.$$

$\square$

**Claim F.31** (2nd term in Eq. (8)). *We have*

$$\sum_{\ell=1}^{L} \|(J_{\ell,t} - J_{\ell,t,t+1}) J_{\ell,t}^\top g_\ell^\star\|_2 \leq \frac{1}{6} \|f_t - y\|_2.$$

*Proof.* For the second term in Eq. (8), for any $\ell \in [L]$, we have

$$\|(J_{\ell,t} - J_{\ell,t,t+1}) J_{\ell,t}^\top g_\ell^\star\|_2 \leq \|J_{\ell,t} - J_{\ell,t,t+1}\| \cdot \|J_{\ell,t}^\top g_\ell^\star\|_2$$

$$= \|J_{\ell,t} - J_{\ell,t,t+1}\| \cdot \|J_{\ell,t}^\top (J_{\ell,t} J_{\ell,t}^\top)^{-1} \cdot \frac{1}{L}(f_t - y)\|_2$$

$$\leq \frac{1}{L} \cdot \|J_{\ell,t} - J_{\ell,t,t+1}\| \cdot \|J_{\ell,t}^\top (J_{\ell,t} J_{\ell,t}^\top)^{-1}\| \cdot \|f_t - y\|_2. \tag{10}$$

We bound these term separately. First,

$$\|J_{\ell,t} - J_{\ell,t,t+1}\| = \left\| \mathsf{J}_\ell(W_t) - \int_0^1 \mathsf{J}_\ell((1-s)W_t + sW_{t+1}) \mathsf{d}s \right\|$$

$$\leq \int_0^1 \|\mathsf{J}_\ell(W_t) - \mathsf{J}_\ell((1-s)W_t + sW_{t+1})\| \, \mathsf{d}s$$

$$\leq \int_0^1 \|\mathsf{J}_\ell(W_t) - \mathsf{J}_\ell(W_0)\| + \|\mathsf{J}_\ell(W_0) - \mathsf{J}_\ell((1-s)W_t + sW_{t+1})\| \, \mathsf{d}s$$

$$\leq \|\mathsf{J}_\ell(W_t) - \mathsf{J}_\ell(W_0)\| + \int_0^1 \|\mathsf{J}_\ell(W_0) - \mathsf{J}_\ell((1-s)W_t + sW_{t+1})\| \, \mathsf{d}s$$

$$\leq \widetilde{O}(n^{1/2}\sqrt{L/m}), \tag{11}$$

where by Fact F.22, we know $\|\mathsf{J}_\ell(W_t) - \mathsf{J}_\ell(W_0)\| = \|J_{W_\ell(t)} - J_{W_\ell(0)}\| \leq \widetilde{O}(n^{1/2}\sqrt{L/m})$, the last inequality is by Lemma F.18. For the second term, we have

$$\|(1-s) \cdot \mathrm{vec}(W_\ell(t)) + s \cdot \mathrm{vec}(W_\ell(t+1)) - \mathrm{vec}(W_\ell(0))\|_2$$
$$\leq (1-s) \cdot \|\mathrm{vec}(W_\ell(t)) - \mathrm{vec}(W_\ell(0))\|_2 + s \cdot \|\mathrm{vec}(W_\ell(t+1)) - \mathrm{vec}(W_\ell(0))\|_2$$
$$= (1-s) \cdot \|W_\ell(t) - W_\ell(0)\|_F + s \cdot \|W_\ell(t+1) - W_\ell(0)\|_F$$
$$\leq O(R).$$

This means the perturbation of $(1-s)W_\ell(t) + sW_\ell(t+1)$ with respect to $W_\ell(0)$ is small, for any $\ell \in [L]$, hence $\|\mathsf{J}_\ell(W_0) - \mathsf{J}_\ell((1-s)W_t + sW_{t+1})\| = \widetilde{O}(n^{1/2}\sqrt{L/m})$.

Furthermore, we have

$$\|J_{\ell,t}^\top (J_{\ell,t} J_{\ell,t}^\top)^{-1}\| = \frac{1}{\sigma_{\min}(J_{\ell,t}^\top)} \le \sqrt{2/\lambda_L}, \tag{12}$$

where the second inequality follows from $\sigma_{\min}(J_{\ell,t}) = \sqrt{\lambda_{\min}(J_{\ell,t} J_{\ell,t}^\top)} \ge \sqrt{\lambda_L/2}$ (see Lemma E.3).

Combining Eq. (10), (11) and (12), we have

$$\sum_{\ell=1}^{L} \|(J_{\ell,t} - J_{\ell,t,t+1}) J_{\ell,t}^\top g^\star\|_2 \le \widetilde{O}(\sqrt{nL/m}) \cdot \lambda_L^{-1/2} \cdot \|f_t - y\|_2$$

$$\le \frac{1}{6}\|f_t - y\|_2, \tag{13}$$

where the last step follows from choice of $m$ (Definition F.25). $\qquad\square$

**Claim F.32** (3rd term in Eq. (8)). *We have*

$$\sum_{\ell=1}^{L} \|(J_{\ell,t} - J_{\ell,t,t+1}) J_{\ell,t}^\top (g_{\ell,t} - g_\ell^\star)\|_2 \le \frac{1}{6}\|f_t - y\|_2$$

*Proof.* We can show

$$\|(J_{\ell,t} - J_{\ell,t,t+1}) J_{\ell,t}^\top (g_{\ell,t} - g_\ell^\star)\|_2 \le \|J_{\ell,t} - J_{\ell,t,t+1}\| \cdot \|J_{\ell,t}^\top\| \cdot \|g_{\ell,t} - g_\ell^\star\|_2. \tag{14}$$

Moreover, one has

$$\frac{\lambda_L}{2}\|g_{\ell,t} - g_\ell^\star\|_2 \le \lambda_{\min}(J_{\ell,t} J_{\ell,t}^\top) \cdot \|g_{\ell,t} - g_\ell^\star\|_2$$

$$\le \|J_{\ell,t} J_{\ell,t}^\top g_{\ell,t} - J_{\ell,t} J_{\ell,t}^\top g_\ell^\star\|_2$$

$$= \|J_{\ell,t} J_{\ell,t}^\top g_{\ell,t} - \frac{1}{L}(f_t - y)\|_2$$

$$\le \frac{\sqrt{\lambda_L/n}}{L} \cdot \|f_t - y\|_2. \tag{15}$$

The first step comes from $\lambda_{\min}(J_{\ell,t} J_{\ell,t}^\top) = \lambda_{\min}(G_{\ell,t}) \ge \lambda_L/2$ (see Lemma E.3) and the last step comes from $g_t$ is an $\epsilon_0$ approximate solution. The fourth step follows from Eq. (15) and the fact that $\|(J_{\ell,t} J_{\ell,t}^\top)^{-1}\| \le 2/\lambda_L$. The last step follows from $g_t$ is an $\epsilon_0$ ($\epsilon_0 \le \sqrt{\lambda_L/n}$) approximate solution to the regression.

Consequently, we have

$$\|(J_{\ell,t} - J_{\ell,t,t+1}) J_{\ell,t}^\top (g_{\ell,t} - g_\ell^\star)\|_2 \le \|J_{\ell,t} - J_{\ell,t,t+1}\| \cdot \|J_{\ell,t}^\top\| \cdot \|g_{\ell,t} - g_\ell^\star\|_2$$

$$\le \widetilde{O}(n^{1/2}\sqrt{L/m}) \cdot \widetilde{O}(n^{1/2}\sqrt{L/m}) \cdot \frac{2}{L\sqrt{n\lambda_L}} \cdot \|f_t - y\|_2$$

$$= \widetilde{O}(\frac{\sqrt{n}}{m}) \frac{2}{\sqrt{\lambda_L}} \cdot \|f_t - y\|_2. \tag{16}$$

The second step follows from Eq. (11) and (15) and the fact that $\|J_{\ell,t}\| \le \widetilde{O}(\sqrt{nL/m})$ (see Lemma F.18).

Finally, we have

$$\sum_{\ell=1}^{L} \|(J_{\ell,t} - J_{\ell,t,t+1}) J_{\ell,t}^\top (g_{\ell,t} - g_\ell^\star)\|_2 \le \widetilde{O}(\frac{\sqrt{nL}}{m}) \frac{2}{\sqrt{\lambda_L}} \cdot \|f_t - y\|_2$$

$$\le \frac{1}{6}\|f_t - y\|_2.$$

The last step follows from choice of $m$ (Definition F.25).

$\square$

**Lemma F.33** (Putting it all together). *We have*

$$\|f_{t+1} - y\|_2 \leq \frac{1}{2}\|f_t - y\|_2. \tag{17}$$

*Proof.* Combining Eq. (8), (9), (13), and (**??**), we have proved the second claim, i.e.,

$$\|f_{t+1} - y\|_2 \leq \frac{1}{2}\|f_t - y\|_2.$$

$\square$

### F.6 WEIGHTS ARE NOT MOVING TOO FAR

**Lemma F.34.** *Let $R$ be chosen as in Definition F.23, then the following holds:*
$$\|W_\ell(t+1) - W_\ell(0)\| \leq R, \quad \forall \ell \in [L].$$

*Proof.* It remains to show that $W_t$ does not move far away from $W_0$. First, we have

$$
\begin{aligned}
\|g_{\ell,t}\|_2 &\leq \|g_\ell^\star\|_2 + \|g_{\ell,t} - g_\ell^\star\|_2 \\
&= \frac{1}{L}\|(J_{\ell,t}J_{\ell,t}^\top)^{-1}(f_t - y)\|_2 + \|g_{\ell,t} - g_\ell^\star\|_2 \\
&\leq \frac{1}{L}\|(J_{\ell,t}J_{\ell,t}^\top)^{-1}\| \cdot \|(f_t - y)\|_2 + \|g_{\ell,t} - g_\ell^\star\|_2 \\
&\leq \frac{2}{L\lambda_L} \cdot \|f_t - y\|_2 + \frac{2}{L\sqrt{n}\lambda_L} \cdot \|f_t - y\|_2 \\
&\lesssim \frac{1}{L\lambda_L} \cdot \|f_t - y\|_2
\end{aligned}
\tag{18}
$$

where the third step follows from Eq. (15) and the last step follows from the obvious fact that $1/\sqrt{n}\lambda_L \leq 1/\lambda_L$.

Then

$$
\begin{aligned}
\|W_\ell(k+1) - W_\ell(k)\| &= \|J_{\ell,k}^\top g_{\ell,k}\| \\
&\leq \|J_{\ell,k}\| \cdot \|g_{\ell,k}\|_2 \\
&\leq \widetilde{O}(\sqrt{nL/m}) \cdot \frac{1}{L\lambda_L} \cdot \|f_k - y\|_2 \\
&\leq \widetilde{O}(\sqrt{n/Lm}) \cdot \frac{1}{2^k\lambda_L} \cdot \|f_0 - y\|_2 \\
&\leq \widetilde{O}(\frac{n}{\sqrt{Lm}}) \cdot \frac{1}{2^k\lambda_L}.
\end{aligned}
$$

The third step uses the fact that $\|J_{\ell,k}\| \leq \|J_{\ell,k}\|_F \leq O(\sqrt{\frac{n}{Lm}})$ by Lemma F.18, and the last step uses the fact that both $\|f_0\|_2$ and $\|y\|_2$ are in the order of $O(1)$.

Consequently, we have

$$
\begin{aligned}
\|W_\ell(t+1) - W_\ell(0)\| &\leq \sum_{k=0}^{t} \|W_\ell(k+1) - W_\ell(k)\| \\
&\leq \sum_{k=0}^{t} \widetilde{O}(\frac{n}{\sqrt{Lm}}) \cdot \frac{1}{\lambda_L}\frac{1}{2^k} \\
&\leq \widetilde{O}(\frac{n}{\sqrt{Lm}}) \cdot \frac{1}{\lambda_L}.
\end{aligned}
$$

By the choice of $R$ (Definition F.23), we know this is upper bounded by $R$. This concludes our proof.

$\square$

# G   PROOF OF LEMMA F.7

In this section, we prove a technical lemma (Lemma F.7) involving truncated gaussian distribution, which correlates to the shifted ReLU activation we use.

**Definition G.1** (Truncated Gaussian distribution). *Suppose $X \sim \mathcal{N}(0, \sigma^2)$. Let $b \in \mathbb{R}$. Then, we say a random variable $Y$ follows from a truncated Gaussian distribution $\mathcal{N}_b(0, \sigma^2)$ if $Y = X | X \geq b$. The probability density function for $\mathcal{N}_b(0, \sigma^2)$ is as follows:*

$$f(y) = \frac{1}{\sigma(1 - \Phi(b/\sigma))} \cdot \frac{1}{\sqrt{2\pi}} e^{-y^2/(2\sigma^2)} \quad y \in [b, \infty),$$

*where $\Phi(\cdot)$ is the standard Gaussian distribution's CDF.*

**Fact G.2** (Properties of truncated Gaussian distribution). *For $b \in \mathbb{R}$, suppose $X \sim \mathcal{N}_b(0, \sigma^2)$. Let $\beta := b/\sigma$. Then, we have*

- $\mathbb{E}[X] = \frac{\sigma \phi(\beta)}{1 - \Phi(\beta)}$, *where $\phi(x) := \frac{1}{\sqrt{2\pi}} e^{-x^2/2}$.*

- $\mathbf{Var}[X] = \sigma^2(1 + \beta\phi(\beta)/(1 - \Phi(\beta)) - (\phi(\beta)/(1 - \Phi(\beta)))^2)$.

- $X/\sigma \sim \mathcal{N}_{b/\sigma}(0, 1)$.

- *When $\sigma = 1$, $X$ is $C(b + 1)$-subgaussian, where $C > 0$ is an absolute constant.*

**Lemma G.3** (Concentration inequality for $b$-truncated chi-square distribution). *For $b \in \mathbb{R}$, $n > 0$, let $X \sim \chi^2_{b,n}$; that is, $X = \sum_{i=1}^{n} Y_i^2$ where $Y_1, \ldots, Y_n \sim \mathcal{N}_b(0, 1)$ are independent $b$-truncated Gaussian random variables. Then, there exist two constants $C_1, C_2$ such that for any $t > 0$,*

$$\Pr\left[\left|X - n(1 + \frac{b\phi(b)}{1 - \Phi(b)})\right| \geq nt\right] \leq \exp\left(-C_1 nt^2/b^4\right) + \exp\left(-C_2 nt/b^2\right).$$

*In particular, we have*

$$\Pr\left[|X - n(1 + b(b + 1))| \geq t\right] \leq \exp\left(-C_1 t^2/(nb^4)\right) + \exp\left(-C_2 t/b^2\right).$$

*Proof.* Since we know that $Y_i \sim \mathcal{N}_b(0, 1)$ is $C(b + 1)$-subgaussian, it implies that $Y_i^2$ is a sub-exponential random variable with parameters $(4\sqrt{2}C^2(b+1)^2, 4C^2(b+1)^2)$. Hence, by the standard concentration of sub-exponential random variables, we have

$$\Pr\left[\left|\sum_{i=1}^{n} Y_i^2 - n\mathbb{E}[Y_i^2]\right| \geq nt\right] \leq \begin{cases} 2\exp\left(-\frac{nt^2}{2 \cdot 32 C^4 (b+1)^4}\right) & \text{if } nt \leq 8C^2(b+1)^2 \\ 2\exp\left(-\frac{nt}{2 \cdot 4 C^2 (b+1)^2}\right) & \text{otherwise} \end{cases}$$

$$\leq 2\exp\left(-C_1 nt^2/b^4\right) + 2\exp\left(-C_2 nt/b^2\right).$$

$\square$

**Fact G.4.** *Let $h \in \mathbb{R}^p$ be fixed vectors and $h \neq 0$, let $b > 0$ be a fixed scalar, $W \in \mathbb{R}^{m \times p}$ be random matrix with i.i.d. entries $W_{i,j} \sim \mathcal{N}(0, \frac{2}{m})$ and vector $v \in \mathbb{R}^m$ defined as $v_i = \phi((Wh)_i) = \mathbf{1}[(Wh)_i \geq b](Wh)_i$. Then*

- *$|v_i|$ follows i.i.d. from the following distribution: with probability $1 - e^{-b^2 m/(4\|h\|^2)}$, $|v_i| = 0$, and with probability $e^{-b^2 m/(4\|h\|^2)}$, $|v_i|$ follows from truncated Gaussian distribution $\mathcal{N}_b(0, \frac{2}{m}\|h\|_2^2)$.*

- *$\frac{m\|v\|_2^2}{2\|h\|_2^2}$ is in distribution identical to $\chi^2_{b',\omega}$ ($b'$-truncated chi-square distribution of order $\omega$) where $\omega$ follows from binomial distribution $\mathcal{B}(m, e^{-b^2 m/(4\|h\|^2)})$ and $b' = \frac{\sqrt{m/2}}{\|h\|_2} b$.*

*Proof.* We assume each vector $W_i$ is generated by first generating a gaussian vector $g \sim \mathcal{N}(0, \frac{2}{m} I)$ and then setting $W_i = \pm g$ where the sign is chosen with half-half probability. Now, $|\langle W_i, h\rangle| = |\langle g, h\rangle|$ only depends on $g$, and is in distribution identical to $\mathcal{N}_b(0, \frac{2}{m}\|h\|_2^2)$. Next, after the sign is

determined, the indicator $\mathbf{1}[(W_i h)_i \geq b]$ is 1 with probability $e^{-b^2 m/(4\|h\|^2)}$ and 0 with probability $1 - e^{-b^2 m/(4\|h\|^2)}$. Therefore, $|v_i|$ satisfies the aforementioned distribution. As for $\|v\|_2^2$, letting $\omega \in \{0, 1, \ldots, m\}$ be the variable indicates how many indicators are 1, then $\omega \sim \mathcal{B}(m, e^{-b^2 m/(4\|h\|^2)})$ and $\frac{m\|v\|_2^2}{2\|h\|_2^2} \sim \chi^2_{b',\omega}$, where $b' = \frac{\sqrt{m/2}}{\|h\|_2} b$. $\qquad\square$

**Fact G.5** (Gaussian tail bound). *For any $b > 0$, we have*

$$\frac{e^{-b^2/2}}{C(b+1)} \leq 1 - \Phi(b) \leq e^{-b^2/2},$$

*where $C$ is an absolute constant.*

We prove a truncated Gaussian version of Lemma 7.1 of Allen-Zhu et al. (2019a).

**Lemma G.6.** *Let $b > 0$ be a fixed scalar. Let the activation function be defined as $\phi(x) := \sqrt{c_b} \mathbf{1}[x > \sqrt{2/m}b] x$, where $c_b := (2(1 - \Phi(b) + b\phi(b)))^{-1/2}$. Let $\epsilon \in (0, 1)$, then over the randomness of $W(0)$, with probability at least $1 - O(nL) \cdot \exp(-\Omega(m \exp(-b^2/2)\epsilon^2/L^2))$, we have*

$$\|h_{i,\ell}\|_2 \in [1 - \epsilon, 1 + \epsilon], \forall i \in [n], \ell \in [L].$$

*Proof.* We only prove for a fixed $i \in [n]$ and $\ell \in \{0, 1, 2, \ldots, L\}$ because we can apply union bound at the end. Below, we drop the subscript $i$ for notational convenience, and write $h_{i,\ell}$ and $x_i$ as $h_\ell$ and $x$ respectively.

According to Fact G.4, fixing any $h_{\ell-1} \neq 0$ and letting $W_\ell$ be the only source of randomness, we have

$$\frac{m}{2}\|h_\ell\|_2^2 \sim \chi^2_{b/\|h\|_2,\omega}, \quad \text{with} \quad \omega \sim \mathcal{B}(m, 1 - \Phi(b')),$$

where $b' := b/\|h_{\ell-1}\|_2$.

We first consider the $\ell = 1$ case. Then, we have $\|h_{\ell-1}\|_2 = 1$, and $b' = b$. Let $P_b := 1 - \Phi(b)$. By Chernoff bound, for any $\delta \in (0, 1)$, we have

$$\Pr[\omega \in (1 \pm \delta)mP_b] \geq 1 - \exp(-\Omega(\delta^2 P_b m)).$$

In the following proof, we condition on this event. By Fact G.5,

$$\omega \in (1 \pm \delta)P_b m \iff \omega \in \left[(1 - \delta)\frac{e^{-b^2/2}}{C(b+1)}m, (1 + \delta)\exp(-b^2/2)m\right].$$

By Lemma G.3, we have

$$\Pr\left[\left|\frac{m}{2}\|h_1\|_2^2 - \omega\left(1 + \frac{b\phi(b)}{P_b}\right)\right| > t\right] \leq \exp\left(-\Omega(t^2/(\omega b^4))\right) + \exp\left(-\Omega(t/b^2)\right)$$

Note that

$$\omega\left(1 + \frac{b\phi(b)}{P_b}\right) \in (1 \pm \delta)mP_b + (1 \pm \delta)mP_b \cdot \frac{b\phi(b)}{P_b} = (1 \pm \delta)(P_b + b\phi(b)) \cdot m.$$

Let $c_b^{-1} := 2(P_b + b\phi(b))$ be the normalization constant. Then, we have

$$\Pr[|c_b\|h_1\|_2^2 - (1 \pm \delta)| > 2tc_b/m] \leq \exp\left(-\Omega(t^2/(\omega b^4))\right) + \exp\left(-\Omega(t/b^2)\right).$$

We want $2tc_b/m = O(\delta)$, i.e., $t = O(\delta c_b^{-1} m)$. Then, we have $\omega t = m^{\Omega(1)} > b^2$. Hence, by Lemma G.3, we actually have

$$\Pr[|c_b\|h_1\|_2^2 - (1 \pm \delta)| > O(\delta)] \leq \exp\left(-\Omega(\delta m/(c_b b^2))\right).$$

By taking $\delta = \epsilon/L$, we get that

$$\|h_1\|_2^2 \in [1 - \epsilon/L, 1 + \epsilon/L]$$

with probability at least

$$1 - \exp(-\Omega(\epsilon^2 P_b m/L^2)) - \exp\left(-\Omega(\epsilon m/(c_b b^2 L))\right) \geq 1 - \exp(-\Omega(\epsilon^2 P_b m/L^2)),$$

where the last step follows from $\frac{1}{c_b b^2} = \frac{P_b + b\phi(b)}{b^2} = \Theta(P_b)$.

We can inductively prove the $\ell > 1$ case. Since the blowup of the norm of $h_1$ is $1 \pm \epsilon/L$, the concentration bound is roughly the same for $h_\ell$ for $\ell \geq 2$. Thus, by carefully choosing the parameters, we can achieve $\|h_\ell\|_2^2 \in [(1 - \epsilon/L)^\ell, (1 + \epsilon/L)^\ell]$ with high probability.

In this end, by a union bound over all the layers $\ell \in [L]$ and all the input data $i \in [n]$, we get that

$$\|h_{i,\ell}\|_2 \in [1 - \epsilon, 1 + \epsilon]$$

with probability at least

$$1 - O(nL)\exp(-\Omega(\epsilon^2 P_b m/L^2)),$$

which completes the proof of the lemma. □

