# OpenReview forum: "Training Multi-Layer Over-Parametrized Neural Network in Subquadratic Time"
_ICLR.cc/2022/Conference — ICLR 2022 Submitted_

### Official Review · Reviewer_xVuq · 2021-10-23

**Correctness:** 4
**Technical Novelty And Significance:** 3
**Empirical Novelty And Significance:** 1
**Recommendation:** 6
**Confidence:** 4

**Main Review:**

This paper is clearly written. The detailed proofs are left in the appendix and the authors explained the high-level proof techniques in the main text. The pros and cons are as follows.

Pros.
1. As far as I know, this paper is the first work that studied the convergence of second-order methods in multi-layer neural networks. Some of the analysis techniques can be useful in future studies.
2. This paper reduced the per-iteration cost of second-order methods by combining the Gram-Gauss-Newton method, tensor-based sketching, and preconditioning. It's a crucial challenge in second-order methods to reduce the per-iteration cost. The ideas in this paper can inspire new algorithms that enjoy lower per-iteration costs.

Cons.
1. The convergence proof of second-order methods seems very similar to the NTK proof for first-order methods. I understand there are some differences in proof details, but I still wonder if there is any conceptual difference in the proof.
2. The linear convergence has been established for first-order methods in similar settings. It seems to me that this paper does not show any benefits of second-order methods over first-order methods in this over-parameterized network setting.
3. The convergence analysis is restricted to the NTK regime. However, in practice, neural networks are not trained in the NTK regime and also achieved better performance in test data than the neural tangent kernel.
4. The reduction of the per-iteration cost relies on the assumption that $m\gg n$, which is hardly true in practice. Therefore, I am concerned about the practicability of the proposed algorithm. It can be helpful if there are empirical experiments that verifies the performance of the algorithms in standard architecture and realistic data set.

**Summary Of The Paper:**

1. This paper proved that the second-order method can minimize the training loss in linea rate on multi-layer over-parameterized neural networks. This analysis relies on the connection between neural tangent kernel and over-parameterized neural networks.
2. This paper also reduced the per-iteration cost of second-order methods to $\widetilde{o}(m^2)$ where $m$ is the hidden layer width, by combining Gram-Gauss-Newton method, tensor-based sketching, and efficient data structures that maintain low-rank representations.
3. This paper also designed an algorithm to efficiently solve a regression problem in which the matrix has its rows being tensor products of two vectors. This algorithm may be of general interest.

**Summary Of The Review:**

In general, I appreciate the technical depth of this work as a theory paper. And I think the techniques for reducing per-iteration cost can be useful in future studies. My main concern is the novelty of the convergence analysis and the practicability of the proposed algorithm. Above all, I think this paper is marginally above the acceptance threshold.

-----------------

Thanks for the response. After reading the response and other reviews, I decided to keep my original evaluation.

---

> ### Author Response · Authors · 2021-11-21
> **Thank you for your comments!**
>
> We thank the reviewer for helpful comments.
>
> Regarding the convergence analysis, we note that while we make use of many quantities that have been used to analyze the convergence of first-order method (GD and SGD) since intuitively, second-order method is to use Hessian (in our case, the Gram matrix) to precondition the gradient. While we are not sure how to quantify ``conceptual difference’’ between our analysis and first-order method, we believe our study is still valuable when one tries to evaluate the performance of second-order methods on training deep over-parametrized networks.
> Regarding the advantage of our method compared to first-order methods such as gradient descent, we point out that it is not clear how to realize gradient descent with a subquadratic cost per iteration, since each weight matrix is of size m x m and one has to multiply at least one $m \times m$ matrix with a length m vector for both forward and backward computations. Without extra assumptions on $W$ being sparse, it seems the $O(m^2)$ time is unavoidable. One might observe that certain subroutines of our method can be exploited to achieve a subquadratic cost per iteration for gradient descent, however, asymptotically the running time is no faster than our second-order method since solving the regression is not the bottleneck for our algorithm. Further, we don’t need to choose and tune the learning rate as in first-order methods, which is a clear advantage. We also want to emphasize that obtaining such an algorithmic result is nontrivial, since it implies that we do not have the runtime budget to read the weight matrix at each iteration. We are required to use a highly effective data structure and representation to proceed the algorithm.
>
> Regarding our analysis relies on the NTK regime, we believe that our argument can be sharpened and avoid using NTK by instead using anti-concentration inequalities. Currently, we are still adapting an NTK-style proof to simplify the overall analysis.
>
> Regarding the practicality of our algorithm, we want to highlight that the main message of our paper is a second-order algorithm to train large and deep networks that achieves subquadratic cost per iteration. Our algorithm draws inspiration from the convex optimization community [1, 2, 3, 4, 5, 6, 10, 11], where we use a carefully-designed data structure to maintain a low rank representation of the change of weights and restart the data structure to amortize the update time. To reduce the cost of solving the regression, we make use of tools from the sketching community [7, 8, 9] to 1). Reduce the cost of forming the Jacobian matrix, 2). Solve the regression via a sketching-based preconditioner. We hope our work opens up the door to use asymptotically efficient second-order methods for training deep neural networks that might further reduce the number of iterations.
>
> [1] Lee and Sidford. Path Finding Methods for Linear Programming: Solving Linear Programs in Õ(vrank) Iterations and Faster Algorithms for Maximum Flow. FOCS 2014.
>
> [2] Cohen, Lee and Song. Solving Linear Program in the Current Matrix Multiplication Time. STOC 2019.
>
> [3] Brand, Lee, Sidford and Song. Solving Tall Dense Linear Programs in Nearly Linear Time. STOC 2020.
>
> [4] Brand, Lee, Liu, Saranuarak, Sidford, Song and Wang. Minimum cost flows, MDPs, and ℓ1-regression in nearly linear time for dense instances. STOC 2021.
>
> [5] Lee, Sidford and Wong. A Faster Cutting Plane Method and its Implications for Combinatorial and Convex Optimization. FOCS 2015.
>
> [6] Jiang, Kathuria, Lee, Padmanabhan and Song. A Faster Interior Point Method for Semidefinite Programming. FOCS 2020.
>
> [7] Clarkson and Woodruff. Low Rank Approximation and Regression in Input Sparsity Time. STOC 2013.
>
> [8] Song, Woodruff and Zhong. Relative Error Tensor Low Rank Approximation. SODA 2019.
>
> [9] Ahle, Kapralov, Knudsen, Pagh, Velingker, Woodruff and Zandieh. Oblivious Sketching of High-Degree Polynomial Kernels. SODA 2020.
>
> [10] Dong, Lee and Ye. A Nearly-Linear Time Algorithm for Linear Programs with Small Treewidth: A Multiscale Representation of Robust Central Path. STOC 2021.
>
> [11] Jiang, Song, Weinstein and Zhang. Faster Dynamic Matrix Inverse for Faster LPs. STOC 2021.

---

### Official Review · Reviewer_aph9 · 2021-11-01

**Correctness:** 2
**Technical Novelty And Significance:** 3
**Empirical Novelty And Significance:** Not applicable
**Recommendation:** 5
**Confidence:** 4

**Main Review:**

1: [motivation and claims about the literature]

> In the first paragraph of introduction, the authors claim that “most theoretical works focus on training on-hidden layer networks”. This is not the case.
>>a), quite a few of the cited papers are theoretical analyses on *deep* neural networks. For example, Jacot et al (2018), Du et al (2019a), and Allen-zhu et al (2019 a, b), which are listed just above this claim in the paper.

>>b), the paper misses several theoretical works on *deep* neural networks. For example, see the references: Lee et al (2019), Liu et al (2020 a,b).
There should be more works on deep neural networks, and I suggest the authors have a more complete query of the literature before making the claims.

> The authors claim in page 1 that “ convergence rate (of first-order methods) is typically slow in non-convex settings …”. I cannot agree with this. For example, in both Du et al (2019a) and Liu et al (2020a), exponential convergence rates on over-parameterized neural networks are obtained for gradient descent methods. Typically, the setting is non-convex, see Liu et al (2020a). For comparison, the proposed method in this paper does not have a faster rate than those two.

> I am not clear how the author got this claim: “it is increasingly evident that first-order methods are becoming a real bottleneck for many practical applications”. The authors should provide evidence.

> The paper claims a running time of O(m^2) for first-order methods, like gradient descent. However, with parallelism this running time can be greatly reduced.

2: [Paper structure]
One uncomfortable thing about this paper is that all the formal statements (e.g., theorems, assumptions) are not shown in the main text, but are deferred to the appendix. Even the formal version of the proposed algorithm is also not displayed in the main pages. In the first 9 pages, there are only informal statements and intuition. I strongly suggest the authors put the main theorems and algorithm, as well as key analysis, in the main text, and just put tedious proof or unimportant details into the appendix.

3: [practicality]
In practical cases, the network width m is never too large, compared to the data size n. In these cases, the proposed algorithm seems to have no advantage compared to gradient descent methods, in terms of the computation cost. Moreover, the complicated design of this algorithm may make it not favorable to practitioners.


References:

Lee, et al. Wide neural networks of any depth evolve as linear models under gradient descent. NeurIPS, 2019.

Liu, et al (a). Toward a theory of optimization for over-parameterized systems of non-linear equations: the lessons of deep learning, Arxiv: 2003.00307, 2020.

Liu, et al (b). On the linearity of large non-linear models: when and why the tangent kernel is constant. NeurIPS, 2020.

**Summary Of The Paper:**

This paper aims to propose a second-order algorithm for training of neural networks with very large widths. The main claim is that the computation cost of each iteration of the proposed methods is sub-quadratic in terms of the network width.

**Summary Of The Review:**

The main text of the paper is not self-containing and not justifiable (suggest to reorganize the paper).
Several claims about the literature are not accurate.
No advantage compared to gradient descent methods for practical scenarios.

---

> ### Author Response · Authors · 2021-11-21
> **Thank you for your comments!**
>
> We thank the reviewer for valuable suggestions. We’ve included the citation of several important works pointed out by the reviewer in our updated pdf.
>
> Regarding the convergence behavior of first-order methods, we point out that the result obtained in [1] has an exponential dependence on the number of layers L on the width of network and convergence rate, while ours only has mild polynomial dependence on L. Compare to [2], we note that their assumptions cannot be extended to ReLU activation, which is the setting studied in our paper. To achieve subquadratic cost per iteration, it is crucial for us to consider ReLU activation since intuitively, it makes sure that after activation, the vector is relatively sparse, so we are not sure the convergence result obtained in [2] can be directly compared to ours.
>
> Regarding the computation cost of gradient descent, we want to highlight that this work focuses on designing a training algorithm that achieves subquadratic cost per iteration while ensuring a good (exponential) convergence rate. From this perspective, we believe the right model to compare our algorithm with is a single-machine version of gradient descent, which would incur an $O(m^2)$ cost per iteration, since the weight matrix is of size $m \times m$ and it is required to multiply it with a length $m$ vector in both forward and backward computation. To some extent, achieving a subquadratic cost per iteration is nontrivial since this requires us not to read the weight matrix at each iteration. We appreciate the reviewer for pointing out an interesting future direction about parallelising our method.
>
> Regarding the organization of this paper, we believe our format can better deliver the contents of our paper, since there are several important components of this paper: 1). Design a low rank maintenance data structure for change of weight matrix, 2). Design a fast sketching-based algorithm to approximate the Jacobian and solve the regression problem, 3). Prove all the errors incurred by approximation do not affect the convergence too much, 4). Prove the convergence of our second-order method. All these parts are self-contained and the individual analysis is long and involved. Hence, we choose to give informal statements and intuitions in the first 9 pages so that readers can better understand our diverse techniques in algorithm design that draw inspirations from second-order method in convex optimization [3, 4, 5, 6, 7, 8, 12, 13] and sketching [9, 10, 11], and in convergence analysis. The same applies to our formal algorithm --- we choose to delay it in the appendix since the formal version might be more than one page long as it utilizes different tools we designed across this paper.
>
> [1] Du, Lee, Li, Wang and Zhai. Gradient Descent Finds Global Minima of Deep Neural Networks. ICML 2019.
>
> [2] Liu, Zhu and Belkin. Loss landscapes and optimization in over-parameterized non-linear systems and neural networks. 2021.
>
> [3] Lee and Sidford. Path Finding Methods for Linear Programming: Solving Linear Programs in Õ(vrank) Iterations and Faster Algorithms for Maximum Flow. FOCS 2014.
>
> [4] Cohen, Lee and Song. Solving Linear Program in the Current Matrix Multiplication Time. STOC 2019.
>
> [5] Brand, Lee, Sidford and Song. Solving Tall Dense Linear Programs in Nearly Linear Time.  STOC 2020.
>
> [6] Brand, Lee, Liu, Saranuarak, Sidford, Song and Wang. Minimum cost flows, MDPs, and ℓ1-regression in nearly linear time for dense instances. STOC 2021.
>
> [7] Lee, Sidford and Wong. A Faster Cutting Plane Method and its Implications for Combinatorial and Convex Optimization. FOCS 2015.
>
> [8] Jiang, Kathuria, Lee, Padmanabhan and Song. A Faster Interior Point Method for Semidefinite Programming. FOCS 2020.
>
> [9] Clarkson and Woodruff. Low Rank Approximation and Regression in Input Sparsity Time. STOC 2013.
>
> [10] Song, Woodruff and Zhong. Relative Error Tensor Low Rank Approximation. SODA 2019.
>
> [11] Ahle, Kapralov, Knudsen, Pagh, Velingker, Woodruff and Zandieh. Oblivious Sketching of High-Degree Polynomial Kernels. SODA 2020.
>
> [12] Dong, Lee and Ye. A Nearly-Linear Time Algorithm for Linear Programs with Small Treewidth: A Multiscale Representation of Robust Central Path. STOC 2021.
>
> [13] Jiang, Song, Weinstein and Zhang. Faster Dynamic Matrix Inverse for Faster LPs. STOC 2021.

---

### Official Review · Reviewer_91V6 · 2021-11-04

**Correctness:** 4
**Technical Novelty And Significance:** 2
**Empirical Novelty And Significance:** Not applicable
**Recommendation:** 6
**Confidence:** 4

**Main Review:**

My first concern is regarding the intuition of this paper. The author claims in the introduction that “it is increasingly evident that first-order methods are becoming a real bottleneck for many practical applications” because the convergence rate is typically slow, i.e., requiring $\mathrm{poly}(n, L, log(1/\epsilon))$ to converge, while second-order method enjoys a much faster convergence rate ($\log(1/\epsilon)$). This is true in terms of the iteration complexity, but it is also true that the per-iteration complexity of the first-order method is much lower (could be lower by $\mathrm{poly}(m)$, while $m$ itself is a high-degree polynomial of $n$), then it is hard to directly claim that first-order method is not as efficient as second-order method.

In order to demonstrate the efficiency of the second-order method, the authors may need to characterize the total computational complexity of both first-order and second-order algorithms, rather than the \textit{cost per iteration}. This could be a more fair comparison to justify the better performance of second-order methods.

It is known that when the data distribution is good, the neural network does not need to be that large to achieve perfect training accuracy. For example, [1,2] show that when the data are well separated by the neural tangent kernel, polylogarithmic widths are sufficient. My suggestion is that the analysis should not be restricted in the setting where $m$ is super large, but consider a wide range of different $m$, and identify under which conditions on $m$, the proposed algorithm could outperform previous ones and vice versa.

1. Ji and Telgarsky, Polylogarithmic width suffices for gradient descent to achieve arbitrarily small test error with shallow ReLU networks. ICLR 2020.
2. Chen et. al., How Much Over-parameterization Is Sufficient to Learn Deep ReLU Networks?. ICLR 2021.


Minor comments:

Could you elaborate on why the update rule in Step 1 can be solved via minimizing the second equation after Step 1?

Can the same algorithm be applied to classification problems, like [3, 4]?

3. Zou et al., Gradient descent optimizes over-parameterized deep ReLU networks. Machine Learning  2019
4. Cao and Gu, Generalization Bounds of Stochastic Gradient Descent for Wide and Deep Neural Networks, NeurIPS 2019



**Summary Of The Paper:**

This paper studies the training algorithms for multi-layer over-parameterized neural networks. In particular, this paper starts from gauss-newton-methods and incorporates tensor-based sketching techniques and preconditioning to improve the per-iteration computational complexity. As a result, the proposed algorithm can find the global solution within the time that is subquadratic in the network width.

**Summary Of The Review:**

This paper is theoretically sound but could be further improved based on my comments.

---

> ### Author Response · Authors · 2021-11-21
> **Thank you for your comments!**
>
> We thank the reviewer for the valuable suggestions.
>
> Regarding the comparison of the cost per iteration of first-order method and second-order method, we notice that the weight matrix of an over-parametrized network is of size $m \times m$, to perform a basic gradient descent update, we need to perform a forward computation, which involves multiplying an $m \times m$ matrix with a length $m$ vector that can take $O(m^2)$ time per layer. Similarly, as we have demonstrated in our paper, computing the gradient also involves matrix-vector products that would incur $O(m^2)$ time per layer. It is not clear that even for gradient descent, how one could achieve a subquadratic cost per iteration. One might notice that certain subroutines of our second-order algorithm can be used to implement a subquadratic gradient descent algorithm. However, we point out that this algorithm is no faster (asymptotically) than our second-order algorithm, further, we do not need to choose and tune learning rate. Hence, we present a second-order algorithm instead of first-order.
>
> In terms of comparing our method under different width settings, we want to point out that the central message of our paper is a new approach to design a second-order algorithm that achieves subquadratic cost per iteration. Inspired by the well-adaptation of second-order methods in convex settings, such as linear programming [1, 2, 3, 4, 10, 11] and semidefinite programming [5, 6], we design novel data structures that can maintain the change of weight matrix and update only when needed. By doing so, we avoid reading the weight matrix at each iteration, rather, we maintain a compact representation of it and show that is enough to implement our algorithm, which is inspired by the sampling idea as in [2]. To improve the running time of computing the update direction (which is equivalent to solve a regression), we leverage ideas from the sketching community [7], where we compute the rows of Jacobian matrix via fast tensor computation [8, 9] and solve the overall regression problem by a sketching-based preconditioner. We wish this work to guide the future second-order algorithm design for training large and deep neural networks.
>
> [1] Lee and Sidford. Path Finding Methods for Linear Programming: Solving Linear Programs in Õ(vrank) Iterations and Faster Algorithms for Maximum Flow. FOCS 2014.
>
> [2] Cohen, Lee and Song. Solving Linear Program in the Current Matrix Multiplication Time. STOC 2019.
>
> [3] Brand, Lee, Sidford and Song. Solving Tall Dense Linear Programs in Nearly Linear Time. STOC 2020.
>
> [4] Brand, Lee, Liu, Saranuarak, Sidford, Song and Wang. Minimum cost flows, MDPs, and ℓ1-regression in nearly linear time for dense instances. STOC 2021.
>
> [5] Lee, Sidford and Wong. A Faster Cutting Plane Method and its Implications for Combinatorial and Convex Optimization. FOCS 2015.
>
> [6] Jiang, Kathuria, Lee, Padmanabhan and Song. A Faster Interior Point Method for Semidefinite Programming. FOCS 2020.
>
> [7] Clarkson and Woodruff. Low Rank Approximation and Regression in Input Sparsity Time. STOC 2013.
>
> [8] Song, Woodruff and Zhong. Relative Error Tensor Low Rank Approximation. SODA 2019.
>
> [9] Ahle, Kapralov, Knudsen, Pagh, Velingker, Woodruff and Zandieh. Oblivious Sketching of High-Degree Polynomial Kernels. SODA 2020.
>
> [10] Dong, Lee and Ye. A Nearly-Linear Time Algorithm for Linear Programs with Small Treewidth: A Multiscale Representation of Robust Central Path. STOC 2021.
>
> [11] Jiang, Song, Weinstein and Zhang. Faster Dynamic Matrix Inverse for Faster LPs. STOC 2021.

---

### Official Review · Reviewer_djYB · 2021-11-08

**Correctness:** 4
**Technical Novelty And Significance:** 3
**Empirical Novelty And Significance:** Not applicable
**Recommendation:** 5
**Confidence:** 3

**Main Review:**

This paper provides an algorithm for training neural networks in the over-parametrized regime. The authors go to great length to show that under good implementation which exploits various structural properties (low rank in particular), the runtime complexity of the algorithm should make it practical. It would be nice to provide empirical validation of the method. The results are not surprising: the gains come from careful attention to these structural properties, and extend Cai et al 2019. I consider this a strength for this paper: providing a precise practical guideline on implementation, while providing theoretical convergence guarantees, is a welcome contribution. The paper's diffusion would of course profit a lot from empirical validation + a code release. Do the authors have such an implementation handy?
I believe this paper is a good contribution to the field -- on the other hand, it would be made much stronger with such an implementation. I'm therefore on the fence on acceptance at this time: I want to encourage authors to provide this implementation.

Comments:
- Can you give the meaning of $m$, $n$, $L$, $\epsilon$ before they are used in the introduction?
- What definition of "over-parametrized" NN are you using? It seems that the rank of the jacobian is at most $n$ where $n$ is the size of the mini-batch, in practice? Is this correct?
- The paper (and in particular Theorem 1.1) uses the L2 error -- is this result extendable for any (convex? smooth?) loss function? Can you explain in the main text why the L2 error and Relu activations are necessary, and where the analysis/algorithm break in the more general case?
- The limits of the method are hardly discussed in the main text.
- It might be easier for the reader to follow if the contributions were given before the informal theorem statements.
- Other practical second order methods have been proposed for neural network optimization. In particular, how does this method compare to AdaHessian? https://arxiv.org/pdf/2006.00719.pdf AdaHessian uses a diagonal approximation to the Hessian. How is it related to K-fac ? https://arxiv.org/pdf/1503.05671.pdf
- This sentence "A smarter way is to instead of using the n × n Gram matrix for neural tangent kernel, however this would still incur a cost of O(nm2) per iteration, since each gradient is an m×m matrix and the Jacobian consists of n such gradients." isn't clear. What are you proposing here?
- There are many typos and sentences which are difficult to follow. e.g. Step 2 paragraph (page 6). "ethnic implications" rather than "ethics implications". It would benefit from an attentive proof-reading.


**Summary Of The Paper:**

This paper proposes a second-order algorithm for training neural networks, in the L2 regression setting. It provides a theoretical analysis of its complexity in the over-parametrized regime. It does not provide empirical validation of the method, or an implementation.

**Summary Of The Review:**

I believe this paper provides good theoretical results on a new second order method for optimizing neural networks in the over-parametrized setting. Content-wise, I'd strongly appreciate empirical validation / implementation. On the formal side, the paper is a bit hard to follow, and contains many typos. It would benefit from careful proof-reading, and correction. I do not recommend acceptation in the current state of the paper, but could be convinced otherwise.

---

> ### Author Response · Authors · 2021-11-21
> **Thank you for your comments!**
>
> We thank the reviewer for valuable suggestions. We have updated the pdf so that many typos have been corrected, we’ve also included the definition of various parameters $(m, n, L, \epsilon)$ before using them in the introduction.
>
> For L2 loss and ReLU: we believe the use of ReLU is necessary, since intuitively, using ReLU makes sure that the activation is sparse, which is one of the key components to design a subquadratic algorithm. In this paper, we focus on L2 loss to simplify the analysis, since our algorithm and its theoretical analysis is already complicated enough. We believe our convergence analysis can be extended to other loss functions, as in [1].
>
> Comparison between our method and AdaHessian/K-fac: we note that papers related to these methods majorly focus on experiments and implementations while lacking a convergence analysis. Though we believe that their method might also achieve a linear convergence rate as ours, it is hard for us to compare our method with theirs directly.
>
> Regarding the experiment/implementation perspective, we want to emphasize the main message of our paper is the design and analysis of the first second-order method that trains deep over-parametrized neural networks that achieve subquadratic cost per iteration. To do so, we utilize tools in other areas, especially using second-order methods to solve convex problems, such as linear programming [2, 3, 4, 5, 11, 12] and semidefinite programming [6, 7]. Specifically, we design novel data structures that maintain a low rank factorization of the change matrix and only update it when needed. We note that this essentially means we don’t need to read the weight matrix at each iteration, but only a succinct representation of certain key components that can facilitate our computation, which is inspired by the breakthrough in linear programming solvers [3]. We also adapt the ideas from the sketching and dimensionality reduction community [8], especially that for tensor-typed inputs [9, 10]. We deploy these tools for two purposes: 1). Approximate the rows of Jacobian in $\tilde O(m)$ time, 2). Solving a regression using a sketch-based preconditioner. We hope that the major takeaway of our paper is a new framework for training large and deep neural networks by designing new tools inspired by the well-studied convex optimization algorithms from the theoretical computer science community.
>
> [1] Zou, Cao, Zhou and Gu. Stochastic Gradient Descent Optimizes Over-parameterized Deep ReLU Networks. ACML 2019.
>
> [2] Lee and Sidford. Path Finding Methods for Linear Programming: Solving Linear Programs in Õ(vrank) Iterations and Faster Algorithms for Maximum Flow. FOCS 2014.
>
> [3] Cohen, Lee and Song. Solving Linear Program in the Current Matrix Multiplication Time. STOC 2019.
>
> [4] Brand, Lee, Sidford and Song. Solving Tall Dense Linear Programs in Nearly Linear Time. STOC 2020.
>
> [5] Brand, Lee, Liu, Saranuarak, Sidford, Song and Wang. Minimum cost flows, MDPs, and ℓ1-regression in nearly linear time for dense instances. STOC 2021.
>
> [6] Lee, Sidford and Wong. A Faster Cutting Plane Method and its Implications for Combinatorial and Convex Optimization. FOCS 2015.
>
> [7] Jiang, Kathuria, Lee, Padmanabhan and Song. A Faster Interior Point Method for Semidefinite Programming. FOCS 2020.
>
> [8] Clarkson and Woodruff. Low Rank Approximation and Regression in Input Sparsity Time. STOC 2013.
>
> [9] Song, Woodruff and Zhong. Relative Error Tensor Low Rank Approximation. SODA 2019.
>
> [10] Ahle, Kapralov, Knudsen, Pagh, Velingker, Woodruff and Zandieh. Oblivious Sketching of High-Degree Polynomial Kernels. SODA 2020.
>
> [11] Dong, Lee and Ye. A Nearly-Linear Time Algorithm for Linear Programs with Small Treewidth: A Multiscale Representation of Robust Central Path. STOC 2021.
>
> [12] Jiang, Song, Weinstein and Zhang. Faster Dynamic Matrix Inverse for Faster LPs. STOC 2021.

---

### Decision · Program_Chairs · 2022-01-20

**Decision:**

Reject

**Comment:**

This paper studies the performance of second-order algorithms on training multi-layers over-parameterized neural networks. The authors propose an algorithm based on the Gram-Gauss-Newton method, tensor-based sketching techniques, and preconditioning to train such a network, whose runtime is subquadratic in the width of the neural network. While some reviewers provide some weak support, none of them are in strong support, even after the author's response. I think one of the reasons is the lack of empirical experiments. Since the main claim of this paper is an efficient second-order algorithm, some experiments are necessary to back up this claim. Unfortunately, the authors did not try to add such an experiment during the rebuttal. I would suggest the authors add such experiments in the revision.